# Non-chelation control in allylations of α-oxy ketones using group-14 allylatranes

Yuya Tsutsui [ID], Kokoro Shiga [ID][1], Akihito Konishi [ID][1,2] ✉ & Makoto Yasuda [ID][1,2] ✉

Stereoselective nucleophilic additions to α-substituted carbonyl compounds are a crucial area of contemporary research in organic chemistry. Of the various advancements in π-facial selectivity in addition reactions of carbonyl compounds, the (polar) Felkin-Anh model and the chelation model are well recognized for accurately explaining the selectivity of the allylic products. For reactions that involve α-oxy carbonyl groups - known for their broad applications in natural-product synthesis and as effective building blocks in organic synthesis - the stereoselective reaction typically follows the chelation model, favoring *syn*-selective addition. In contrast to the well-established *syn*-selective additions of α-oxy carbonyls, *anti*-selective additions through a non-chelation pathway remain largely unexplored. In this study, we present the *anti*-selective allylation of α-oxy ketones using allylatranes that feature a highly coordinated group-14-element center. These atranes demonstrate high nucleophilicity and low chelating ability due to their transannular interactions and rigid framework, facilitating *anti*-selective allylations. A combined experimental and theoretical approach has been used to highlight the unique electronic properties of these atranes. This method is applicable to a wide variety of substrates, producing *anti*-1,2-diols with a homoallylic moiety in high yield and excellent diastereoselectivity compared to traditional methods.

Controlling the π-facial selectivity in addition reactions to carbonyl compounds is of crucial importance in stereoselective organic synthesis[1-4]. A common case is that addition reactions to carbonyl compounds with a substituent at the α-position results in diastereomers. This stereochemical outcome is accurately explained by the Felkin-Anh model[5,6], which illustrates the relationship between the conformation around the α-carbon and the attacking nucleophile. The nucleophile reacts from the direction opposite to the largest α-carbon substituent ($R_L$), and when the nucleophile approaches the carbonyl carbon at the Bürgi-Dunitz angle[7], $R_L$ avoids steric repulsion orienting away from it, thus avoiding steric repulsion with the medium-sized substituent ($R_M$), favoring a transition state with the nucleophile closer to the smallest substituent ($R_s$) (Fig. 1ai). If there is an electronegative atom (X) at the α-position, the conformation of the transition state is changed. The C–X bond (X = halogen, N, or S) adopts an orthogonal conformation to the carbonyl group due to the stabilizing effect of the hyperconjugation between the $\sigma^*_{C-X}$ and $\pi^*_{C=O}$ orbitals (polar Felkin−Anh Model)[8-11]. Thus, the nucleophile approaches from the direction opposite to the X substituent (Fig. 1aii left). On the other hand, when a chelating Lewis acid (LA)[12] or hydrogen bond[13,14] are present with either α-halogenated carbonyls or α-amino carbonyls, the reaction proceeds via the chelation model[15-19], where the stereoselectivity is the opposite to that of the Felkin-Anh Model. Here, the conformation where the LA chelates with the coordinating electronegative group (X) and the carbonyl oxygen is preferred, and the nucleophile approaches in a manner that avoids steric repulsion with $R_L$ (Fig. 1aii right). For reactions involving α-oxy carbonyl groups, which have broad applications in natural-product synthesis and are effective building blocks in organic

[1]Department of Applied Chemistry, Graduate School of Engineering, The University of Osaka, 2-1 Yamadaoka, Suita, Osaka, Japan. [2]Innovative Catalysis Science Division, Institute for Open and Transdisciplinary Research Initiatives (ICS-OTRI), The University of Osaka, Suita, Osaka, Japan. ✉e-mail: a-koni@chem.eng.osaka-u.ac.jp; yasuda@chem.eng.osaka-u.ac.jp

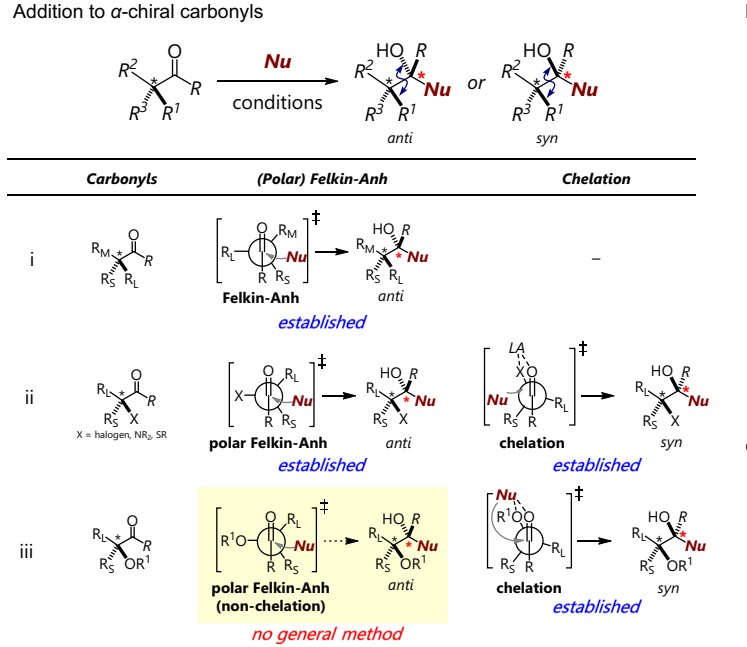

**Fig. 1 | Overview for the diastereoselective additions to α-chiral carbonyls and allylations using atrane-type reagent. a** Summary for the diastereoselective addition to α-chiral carbonyls. **b** anti-Selective allylation of α-oxy ketones by 1E(allyl) (E = Si, Ge, or Sn; this work). **c** Calculated electronic properties of 1Si(allyl) (left) and allylSiPh₃ **10** (right). **d** DFT-supported plausible reaction mechanism for the anti-selective allylation using 1Si(allyl).

synthesis, the stereoselective reaction following the chelation model is strongly effective (Fig. 1a-iii right). One of the diastereoselectivities can be easily controlled because the oxygen groups, which are highly coordinating, readily chelate to nucleophiles. However, obtaining stereoselective outcomes according to the Felkin-Anh model with α-oxy carbonyl compounds has been virtually impossible, except in a few limited cases[2,10], and has remained a long-standing challenge (Fig. 1b). Non-chelation control in the nucleophilic addition to α-oxy carbonyls is the final challenge to be solved in the Felkin-Anh model for the addition of nucleophiles to carbonyl compounds.

The allylation of α-oxy carbonyls frequently follows the Chelation model, yielding syn products where the vicinal hydroxyl groups are positioned on the same side relative to the carbon-chain backbone[20]. When using Sn-[21,22], In-[23,24], Ga-[25], Si-[26,27], and B[28]-based allyl nucleophiles, the rigid conformation of the transition state provides syn products. Another approach involves the use of chelating Lewis acids, such as TiCl₄[29–31], SnCl₄[30,31], or a combined InCl₃−chlorosilane[32]. However, anti-selective allylation via a non-chelation pathway is still relatively unexplored. In a pioneering study, Reetz et al. have reported the anti-selective allylation of α-oxy aldehydes through non-chelation control using allylSiMe₃ and a monodentate Lewis acid (BF₃·Et₂O)[33]. Still, examples of anti-selective allylations are limited and there is room for significant further improvement. Establishing anti-selective stereocontrol in allylations of α-oxy ketones is particularly challenging, considering that ketones are less electrophilic than aldehydes and therefore require a stronger nucleophile for addition. However, enhanced nucleophilicity increases the Lewis acidity of the allyl metal, thus promoting chelation. Achieving anti-selective allylations requires balancing low chelation ability with high nucleophilicity, but the trade-off between these properties hampers the realization of this goal. Using highly coordinated nucleophiles[34] is one of the most promising solutions to realize the allylation of α-oxy ketones under non-chelation control. Our group has previously investigated group-14-element atrane-type cationic species 1E⁺ (E = Si, Ge, or Sn)[35]. The atrane framework contains a highly coordinated element center due to the

intramolecular transannular N−E bond, and thus exhibits significantly enhanced nucleophilicity. The rigid triaryl atrane structure also reduces undesired chelation effects. Thus, we designed allylatranes 1E(allyl) (E = Si, Ge, or Sn) for the anti-selective allylation of α-oxy ketones (Fig. 1b). The synthesis of 1E(allyl) is summarized in Fig. S1.

## Results

As a model reaction, we performed the allylation of benzoin methyl ether **2a** (Table 1). When allylSiMe₃ **4** was used in the presence of BF₃·Et₂O, syn-product **3a** was obtained. The syn-selectivity may be due to either the generation of a high-coordinated or a cationic Si species[36] the applied reaction conditions. The low yield (6%) of this reaction is caused by the low nucleophilicity of **4** (Table 1, entry 1). Oxygen-ligated allylsilatrane allylSi(OC₂H₄)₃N **5**, which exhibits heightened nucleophilicity[37,38], was an ineffective allylation reagent (Table 1, entry 2). Nuclear-magnetic-resonance (NMR) measurements revealed that **5** decomposes rapidly in the presence of BF₃·Et₂O (Fig. S10). Allyl-tributylstannane allylSnBu₃ **6** furnished the chelation-controlled product syn-**3a** in 99% yield with high diastereoselectivity (syn/anti = 99/1; Table 1, entry 3). Here, stannyl cation species generated by transmetalation with BF₃·Et₂O likely activate the substrate in a chelation-controlled manner[39,40]. Treating **2a** with allylSnBu₃/SnCl₂[41] or allylindium[23,24,42] provided product syn-**3a** in high yield with excellent syn-selectivity (syn/anti = >99/1; Table 1, entries 4 and 5) via the chelation pathway. The reactions with allylMgBr **8** or allylLi **9** proceeded without diastereoselectivity due to strong chelation combined with too high nucleophilicity (Table 1, entries 6 and 7)[43,44]. However, we were pleased to find that the desired homoallylic alcohol (**3a**) was produced using newly synthesized **1**Si(allyl) in 97% yield with high anti-selectivity (syn/anti = 5/95; Table 1, entry 8). The stereochemistry of the allylated product anti-**3a** was confirmed using single-crystal X-ray diffraction analysis after silylation of the hydroxy group. Allylgermatrane **1**Ge(allyl) also furnished anti-**3a** with nearly perfect selectivity (syn/anti = 5/95; Table 1, entry 9). A decrease in diastereoselectivity and yield was observed with **1**Sn(allyl) (syn/anti = 14/86, 60%; Table 1, entry

## Table 1 | Allylation of benzoin methyl ether 2a

| Entry | Reagent (equiv.) | Conditions | Yield/% syn/anti | Cause of the failure to obtain anti-3a |
|---|---|---|---|---|
| 1 | 4 / SiMe₃ (1.0) ... BF₃·Et₂O (1.0) | CH₂Cl₂, rt | 6% >99 / 1 | low nucleophilicity of 4 |
| 2 | 5 (1.0) ... BF₃·Et₂O (1.0) | CH₂Cl₂, rt | 0% | decomposition of 5 by BF₃·Et₂O |
| 3 | 6 / SnBu₃ (1.0) ... BF₃·Et₂O (1.0) | CH₂Cl₂, rt | 99% 99 / 1 | transmetalation leading to chelation |
| 4 | 6 / SnBu₃ (1.2) ... SnCl₂ (1.2) | CH₃CN, rt | 100% >99 / 1 | transmetalation leading to chelation |
| 5 | Br (1.5) + In (1.5) → 7 / In | THF, rt | 94% >99 / 1 | chelation by 7 |
| 6 | 8 / MgBr (1.5) | THF, -78 °C to rt | 94% 71 / 29 | chelation by 8 nucleophilicity of 8 too high |
| 7 | 9 / Li (1.5) | THF, -78 °C to rt | 99% 86 / 14 | chelation by 9 nucleophilicity of 8 too high |
| 8 | 1Si(allyl) (1.0) ... BF₃·Et₂O (1.0) | CH₂Cl₂, rt | 97% 5 / 95 | |
| 9 | 1Ge(allyl) (1.0) ... BF₃·Et₂O (1.0) | CH₂Cl₂, rt | 98% 5 / 95 | |
| 10 | 1Sn(allyl) (1.0) ... BF₃·Et₂O (1.0) | CH₂Cl₂, rt | 60% 16 / 84 | |

Yields and diastereomeric ratios of the products were determined by ¹H NMR measurements using 1,1,2,2-tetrachloroethane as an internal standard. Bold numbers represent compound numbers

10), because the nucleophilicity of **1**Sn(allyl) was too high. The high stability of **1**E(allyl) with $BF_3 \cdot Et_2O$ is likely due to the relatively strong $E-C_{Ar}$ bonds and because the chelate effect of the cage-shaped structure suppresses decomposition. Other allyl nucleophiles were not effective in this *anti*-selective allylation (Table S2).

To investigate the electronic properties of **1**E(allyl), we calculated its molecular orbitals (Fig. 1c). The calculated energy of the HOMO of **1**Si(allyl) (−6.18 eV) suggests that its nucleophilicity is higher than that of allylSiPh$_3$ **10** (−6.60 eV). The high nucleophilicity of **1**Si(allyl) is also supported by $^{13}C$ NMR and natural-bond-orbital (NBO) analyses (Table S6). The difference in NBO charge between $C_\beta$ and $C_\gamma$ was higher for **1**Si(allyl) ($|q(C_\beta)-q(C_\gamma)| = 0.277$) than for **10** ($|q(C_\beta)-q(C_\gamma)| = 0.254$). Meanwhile, a second-order-perturbation analysis indicated that the hyperconjugation from the $\sigma(Si-C_\alpha)$ bond to the $\pi^*(C_\beta=C_\gamma)$ bond is less effective in **1**Si(allyl) ($\Delta E = 3.84$ kcal/mol) than in **10** ($\Delta E = 5.61$ kcal/mol), commensurate with weak electronic communication between the allylic moiety and the group-14-element center of **1**Si(allyl). Thus, the strong nucleophilic character of **1**Si(allyl) should reflect the charge localization rather than the stereoelectronic effects. Notably, the results of the NBO analyses indicate that the nucleophilicity of **1**Si(allyl) is similar to that of allylsilatrane **5**, which rapidly decomposes under the applied reaction conditions. This highlights the balance of reactivity and stability in **1**Si(allyl), facilitated by its rigid triaryl-based structure. The stabilization of the silyl center by transannular interactions would also enhance the allyl-anion character. A crystallographic analysis confirmed that the molecular geometry of **1**Si(allyl) exhibits a nearly trigonal-bipyramidal structure with a five-coordinated silicon center. Most allylic reagents with high nucleophilicity feature a strong Lewis-acidic metal center, which facilitates the allylation of *α*-oxy ketones through a chelation pathway, resulting in *syn*-products[9,31,33]. In contrast to the traditional methods for achieving *syn*-selective allylations, **1**E(allyl) enhances the nucleophilicity of the allylic group while simultaneously reducing the Lewis acidity of the metal center. This careful balance between high nucleophilicity and weak Lewis acidity in **1**E(allyl), achieved through an atrane-type framework, allows for the distinctive *anti*-selective allylations of *α*-oxy ketones.

DFT calculations at the B3LYP-D3/6-31 G*/SMD (dichloromethane)//B3LYP-D3/6-31 G* level support the observed *anti*-selectivity via a non-chelation pathway (Fig. S16). A plausible mechanism for the allylation of **2a** with allylsilatrane **1**Si(allyl) is shown in Fig. 1d. First, ketone **2a** coordinates to $BF_3$. The activated ketone then interacts with **1**Si(allyl), forming a carbon−carbon bond. The estimated transition state leading to the *anti*-product is by 6.4 kcal/mol more stable than that leading to the corresponding *syn*-product. This enhanced stability results from steric repulsion between the atrane moiety of **1**Si(allyl) and the phenyl group of **2a**. Interestingly, the conformation in the transition state resembles the Cram-type mode, in which the *α*-methoxy group is located antiperiplanar relative to the carbonyl group of **2a** activated by $BF_3$, rather than following the polar Felkin−Ahn model. The allylation of **2a** with the bulky **1**Si(allyl) and the small Lewis acid $BF_3$ is therefore likely to favor the Cram-type conformation over the initially anticipated polar Felkin−Ahn conformation. Subsequently, one of the fluorine atoms on the boron atom is captured by silyl cation **1**Si$^+$, which results in the formation of the allyl product Int and **1**SiF. NMR monitoring confirmed the generation of **Int** and **1**SiF (Fig. S12). Finally, protonation by quenching yields the homoallylic alcohol *anti*-**3a**. There is a possibility that an in-situ exchange of the allyl moiety between **1**Si(allyl) and $BF_3 \cdot Et_2O$ occurs to form allylBF$_2 \cdot Et_2O$, albeit that a signal for allylBF$_2 \cdot Et_2O$ was not observed in the NMR experiments. Therefore, it is considered that this *anti*-selective allylation proceeds via a non-chelation pathway. Further theoretical calculations to investigate the pathway in detail are currently in progress in our group. The high stability of **1**Si(allyl) derived from the atrane structure suppresses the undesired in-situ exchange process.

The *anti*-product was obtained with high selectivity in high yield from a variety of *α*-oxy ketones using **1**Si(allyl) (conditions A, Table 2). The stereochemistry of *anti*-**3** was determined by comparison with the NMR spectrum of *syn*-**3**, which was prepared using our reported method employing Sn(II) chloride for chelation-control (conditions B, Table 2)[41]. The reactions conducted under conditions A with *α*-iso-propoxy (**2b**), *α*-phenoxy (**2c**), or *α*-acetoxy (**2d**) ketones afforded products **3b**−**d** with high *anti*-selectivity (*syn/anti* = 1/ > 99) (conditions A, Table 2, entries 1 − 3). Under chelation control, the *syn*-products were obtained with perfect selectivity (conditions B, Table 2, entries 1 − 3). A bulky *α*-silyloxy-substituted ketone (**2e**) gave the desired product (**3e**) with high diastereoselectivity (*syn/anti* = 4/96) in 95% yield (conditions A, Table 2, entry 4). The selectivity using **1**Si(allyl) is better than that obtained using allylMgBr (syn/anti = 15/85) as reported by Woerpel[45]. In the case of the Sn(II) system, the selectivity was slightly lower, but the *syn*-product was still obtained with high selectivity (*syn/anti* = 87/13, conditions B, Table 2, entry 4). The *α*-aminoxy-substituted ketone (**2f**) gave the *anti*-product in 98% yield with great *anti*-selectivity (*syn/anti* = 1/ > 99, conditions A, Table 2, entry 5). The reaction also gave good yields and diastereoselectivities with *p*-chloro- or *p*-phenyl-substituted benzoin methyl ethers (**2g**, **2h**) and a naphthyl ketone derivative (**2i**) (conditions A, Table 2, entries 6−8). Furthermore, the allylations of *α*-methyl-*α*-methoxy ketones **2j** and **2k** with a 2,5-disubstituted *trans*-tetrahydrofuran motif afforded **3j** and **3k** with nearly perfect *anti*-selectivity (*syn/anti* = 4/96 and 2/98) (conditions A, Table 2, entries 9−10). The selectivity of **3k** was dramatically improved relative to a previous study with allylMgCl (*syn/anti* = 29/71)[46]. Unfortunately, the desired product was not obtained from the allylation of *α*-oxy aliphatic ketone **2l** (conditions A, Table 2, entry 11), as the allylation of aliphatic ketones requires an allyl nucleophile with higher nucleophilicity; using **1**Sn(allyl) instead of **1**Si(allyl) had no effect. The allylation of *α*-oxy aldehyde **2m** proceeded with low diastereoselectivity (*syn/anti* = 36/64, conditions A, Table 2, entry 12). The nucleophilicity of **1**Si(allyl) toward *α*-oxy aldehydes is assumed to be too high. The allylsilatrane derivatives **1**Si(methallyl) and **1**Si(2-phenylallyl) gave the desired adducts with nearly perfect diastereoselectivity. Methallylsilatrane **1**Si(methallyl) selectively afforded *anti*-**3aa** (*syn/anti* = 7/93) in 78% yield (conditions A, Table 2, entry 13). The allylation with **1**Si(2-phenylallyl) also furnished *anti*-**3ab** with high selectivity (*syn/anti* = 1/99) in 69% yield (conditions A, Table 2, entry 14). It was also confirmed that the *syn*-products could be obtained from the aforementioned examined substrates using the corresponding allylstannane derivatives and Sn(II) chloride. Thus, the use of **1**Si(allyl) (via a non-chelation path) and the Sn(II) system (via a chelation path) affords a high degree of diastereocontrol in the allylations of *α*-oxy ketones.

Then, the reactivity of allylsilatrane **1**Si(allyl) with cyclic *α*-oxy ketones was explored (Fig. 2a). In the case of 2-methoxycyclohexanone **2n**, low diastereoselectivity was observed (*cis/trans* = 62/38), probably due to the flexible conformational orientation of the methoxy group. In contrast, the allylations of conformationally fixed cyclohexanones **2o** and **2p** with a $^t$Bu group at the 4-position, selectively yielded diastereomers via an equatorial attack. For the allylation of **2p**, the product **3p** showed the *trans*-configuration, which is identical to that generated under the reported chelation conditions involving a twist-boat conformation in the transition state[47]. These results indicate that **1**Si(allyl) acts as a bulky allylic nucleophile, favoring nucleophilic attack from the less hindered equatorial position under our non-chelation conditions. The products obtained using **1**Si(allyl), i.e., *cis*-**3o** and *trans*-**3p**, were identical to those reported by Paquette using allylindium[48]. Paquette has proposed a transition state with a twist-boat conformation to maximize the O−In interactions, whereas our method involves a simple equatorial attack. Furthermore, the steric effect of **1**Si(allyl) was evident in the allylation of *α*-oxy cyclobutanone **2q** (Fig. 2b). The resulting allylated product (**3q**) is an important synthetic intermediate

## Table 2 | Substrate scope of α-oxy carbonyls in diastereoselective allylations

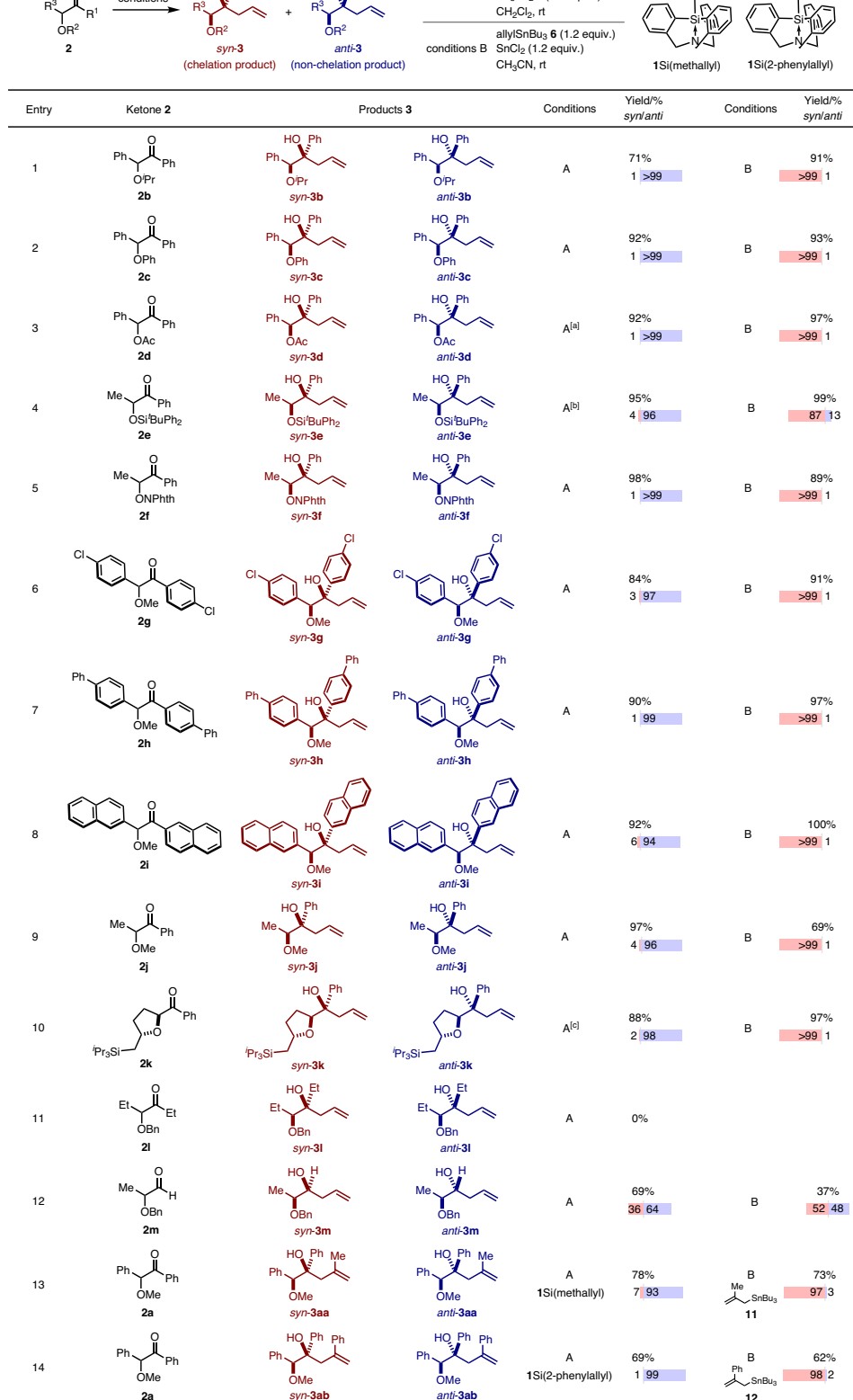

| Entry | Ketone 2 | Products 3 | | Conditions | Yield/% syn/anti | Conditions | Yield/% syn/anti |
|---|---|---|---|---|---|---|---|
| 1 | 2b | syn-3b | anti-3b | A | 71% 1 >99 | B | 91% >99 1 |
| 2 | 2c | syn-3c | anti-3c | A | 92% 1 >99 | B | 93% >99 1 |
| 3 | 2d | syn-3d | anti-3d | A[a] | 92% 1 >99 | B | 97% >99 1 |
| 4 | 2e | syn-3e | anti-3e | A[b] | 95% 4 96 | B | 99% 87 13 |
| 5 | 2f | syn-3f | anti-3f | A | 98% 1 >99 | B | 89% >99 1 |
| 6 | 2g | syn-3g | anti-3g | A | 84% 3 97 | B | 91% >99 1 |
| 7 | 2h | syn-3h | anti-3h | A | 90% 1 99 | B | 97% >99 1 |
| 8 | 2i | syn-3i | anti-3i | A | 92% 6 94 | B | 100% >99 1 |
| 9 | 2j | syn-3j | anti-3j | A | 97% 4 96 | B | 69% >99 1 |
| 10 | 2k | syn-3k | anti-3k | A[c] | 88% 2 98 | B | 97% >99 1 |
| 11 | 2l | syn-3l | anti-3l | A | 0% | | |
| 12 | 2m | syn-3m | anti-3m | A | 69% 36 64 | B | 37% 52 48 |
| 13 | 2a | syn-3aa | anti-3aa | A 1Si(methallyl) | 78% 7 93 | B 11 | 73% 97 3 |
| 14 | 2a | syn-3ab | anti-3ab | A 1Si(2-phenylallyl) | 69% 1 99 | B 12 | 62% 98 2 |

*Conditions A*: **1**Si(allyl) (1.0 eq.), BF₃·Et₂O (1.0 eq.), and CH₂Cl₂ at room temperature. *Conditions B*: allylSnBu₃ **6** (1.2 eq.), SnCl₂ (1.2 eq.), and CH₃CN at room temperature. Yields and diastereomeric ratios of the products were determined using ¹H NMR measurements using 1,1,2,2-tetrachloroethane as an internal standard. [a] Reaction was conducted at 0 °C. [b] **1**Si(allyl) (1.5 equiv.). [c] Reaction was conducted at −20 °C. Bold numbers represent compound numbers.

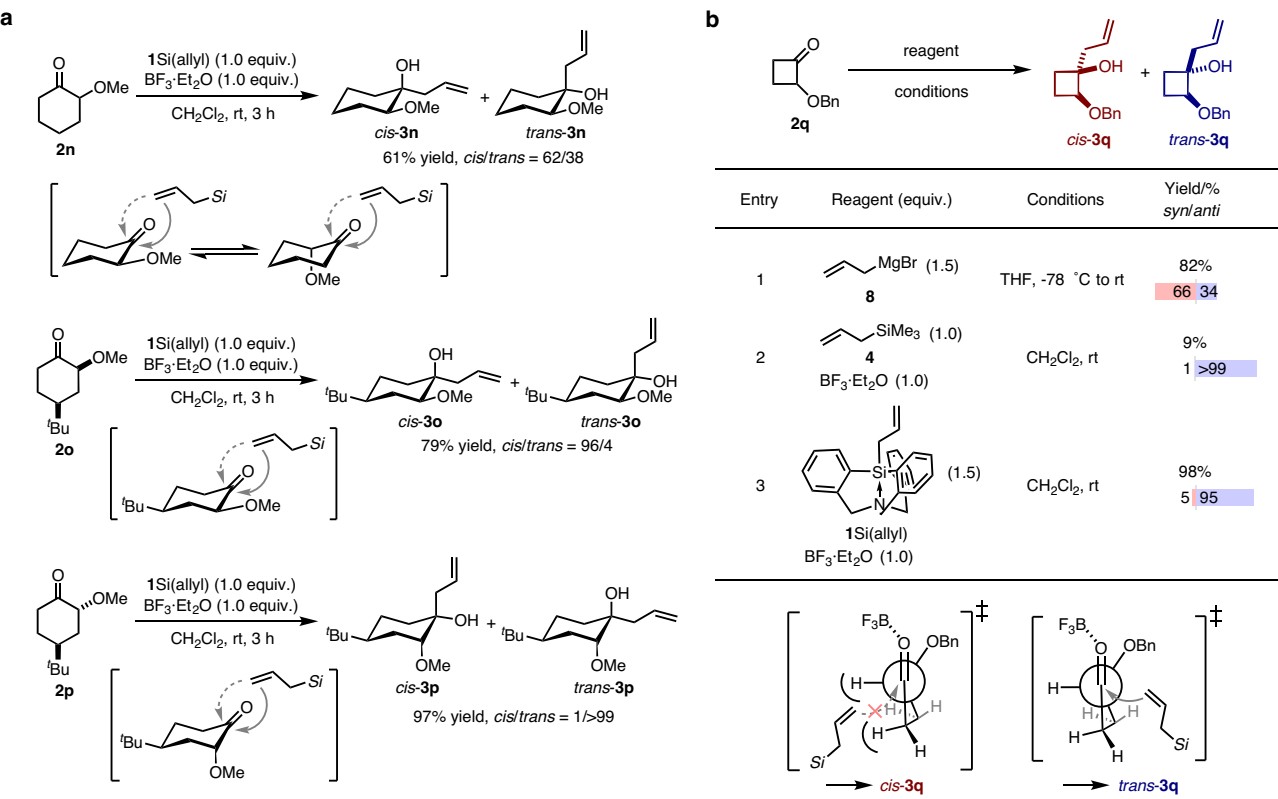

**Fig. 2 | Diastereoselective additions to α-oxy cyclic carbonyls. a** Allylation of α-oxy cyclohexanones **2n–2p**. **b** Allylation of α-oxy cyclobutanone **2q**.

for the substructure of a β-secretase modulator candidate. When **2q** was treated with allylMgBr **8**, the diastereoselectivity was low (*cis/trans* = 66/34; Fig. 2b, entry 1)[49]. Treatment with allylSiMe₃ **4** in the presence of BF₃·Et₂O gave the desired product (*trans*-**3q**) with high diastereoselectivity (*cis/trans* = 1/ > 99; Fig. 2, entry 2), albeit that the yield was disappointing. Silatrane **1**Si(allyl) afforded *trans*-**3q** in high yield and high diastereoselectivity (*cis/trans* = 5/95). The stereochemistry of **3q** was determined by acetalization of the diols obtained from **3q** (Fig. S9). Cyclobutanone **2q** has a rigid conformation due to its four-membered ring and a transition state that minimizes steric repulsion between the hydrogens on the cyclobutane ring and **1**Si(allyl) would be favored. Allylsilatrane **1**Si(allyl) is expected to demonstrate efficacy in natural-product synthesis and pharmaceutical synthesis.

## Methods

### General procedure for *anti*-selective allylations using 1Si(allyl)

In a nitrogen-filled glovebox, a mixture of BF₃ · Et₂O (0.2 mmol) and ketone **2** (0.2 mmol) in dichloromethane (2 mL) was treated with **1**Si(allyl) (0.2 mmol). After the reaction mixture was stirred for 3 h at room temperature, methanol (2 mL) was added to the mixture. All solvents were removed under reduced pressure to give the crude product.

## Data availability

Experimental details, characterization of the compounds, spectral data, and theoretical-calculation results are available within the published manuscript and supplementary information. The X-ray crystallographic data for structures reported in this study have been deposited at the Cambridge Crystallographic Data Center (CCDC) under deposition numbers CCDC 2443722 (for **1**Si(allyl)), 2443723 (for **1**Ge(allyl)), 2443724 (for **1**Sn(allyl)), 2443725 (for **1**Si(methallyl)), 2443726 (for **1**Si(2-phenylallyl)), 2443727 (for *syn*-**3a**), 2443728 (for *anti*-**3a′**), and 2443729 (for *anti*-**3d**). These data can be obtained free of charge from the Cambridge Crystallographic Data Center via www. ccdc.cam.ac.uk/data_request/cif. Source data are provided with this paper. All data are available from the corresponding author upon request. Source data are provided with this paper.

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

## Acknowledgements

This work was financially supported by the MEXT Grant-in-Aid for Transformative Research Areas (A) "Digitalization-driven Transformative Organic Synthesis (Digi-TOS)" (JP21H05212 [MY]), by JST CREST (JPMJCR2OR3 [MY]), and by the Japan Society for the Promotion of Science (JP25K22182 [MY] and JP23H01950 [AK]). A.K. would like to express his gratitude for a "Condensed Conjugation" grant (JP23H04028 [AK]). Part of this work was supported by the JST FOREST Program (under grant number JPMJFR2426 [AK]). The authors acknowledge the Analytical Instrumentation Facility, Graduate School of Engineering, Osaka University.

## Author contributions

M.Y. and A.K. conceived and designed the experiments. Y.T. and K.S. performed the experiments. Y.T. and A.K. carried out the DFT calculations. A.K. performed the crystallographic analyses. Y.T., A.K., and M.Y. co-wrote the manuscript. All authors discussed the results and commented on the manuscript.

## Competing interests

The authors declare no competing interests.
