## [Transparent Peer Review file · Nature Communications]

Non-Chelation Control in Allylations of α -Oxy Ketones Using Group-14 Allylatranes

Corresponding Author: Professor Makoto Yasuda

Version 1:

Reviewer comments:

Reviewer #1

(Remarks to the Author)

Key results

This paper by Tsutsui, Konishi, and Yasuda details the selective anti-allylation of alpha-oxyketones using a bulky (and interesting) allylating agent and a monodentate Lewis acid (BF₃-OEt₂). The reaction proceeds via an open transition state, rendering the stereoselectivity adherent to the traditional Felkin-Anh model and resulting in the anti-product. This reaction is notable for the researchers identifying a solution to having to balance the nucleophilicity of the nucleophile with the Lewis acidity of the nucleophile-carrier, i.e. the atrane is not Lewis acidic at all (precluding a closed transition state, which would provide the syn product) yet the allyl group on the atrane displays enhanced nucleophilicity. The significance of this paper is multifold. On its simplest level, this reaction provides access to a structural motif that has not been readily accessible to-date. At a deeper level, it also provides a blueprint for further development of anti-selective addition reactions of alpha-oxyketones (and potentially related substrates). Finally, it further demonstrates atranes as powerful non-coordinating atom-transfer agents, an idea that has been explored previously by this group and others.

Validity

I have no concerns about the validity or robustness of the data interpretation or conclusions. I appreciate the crystal structures to provide confidence in the interpretation of the stereochemistry of the products, and extrapolation then to the NMR data (using both shifts and splitting) to characterize further products. I really enjoyed the mechanistic studies starting on page S78, highlighting the importance of order of addition in the reaction. I wish there was room to incorporate these studies into the main text.

Significance

On the one hand, the concept that one could pair a Lewis acid with a non-coordinating nucleophile to get addition to alpha-oxyketones via an open transition state and thus anti addition is not surprising. On the other hand, the Felkin-Anh model has existed for more than 50 years and this reaction hasn't previously been well-established. So the devil is in the details of what these researchers have discovered...and that devil is that these allylatranes are unique in their reactivity. So, the real significance is the continued development of atranes as versatile and tunable atom-transfer reagents. I think that is highly significant conceptually, and I believe that the authors know and believe this (given their application of the scaffold previously), though it's not a glaringly obviously important takeaway from the paper. It may be worth just adding some emphasis encouraging other synthetic organic chemists to look at atranes for this type of use...just a bit more on the potential downstream impact of these molecules up front could go a long way. Or even just sprucing up the part of the Figure 1b that highlights the unique properties of the atrane.

I also discuss some matters relating to significance below, in "Suggested Improvements".

Data and methodology

Data and methodology are great, see "Validity" above.

Analytical approach

See "Validity" above – the compounds have been well-characterized, and the conclusions are supported.

Suggested improvements

The experimental work presented certainly supports the claims of the paper. There is the question of how the synthesized products might directly be useful, to which there is really no answer provided. Could one find evidence of such scaffolds in, for example, a bioactive molecule? A chiral ligand? I think it would strengthen the case for significance to see some of these products being transformed into useful molecules – and if that's not easily doable (since these motifs have been difficult to make) then perhaps one could target stereoisomers of important compounds that are known? I think it would help non-synthetic-organic-chemistry readers (who don't know or care about novel exploitation of Felkin-Anh) to understand why this is important.

Further, I found the article stimulating, so it had me thinking about some of the results and ways to explore further... I am not suggesting these are requisite experiments, but rather ideas on the subject:

1. I wonder why the TMS-allyl group give syn selectivity? The low nucleophilicity results in low yield (as mentioned), but doesn't explain the observed diastereoselectivity. Is it possible that the TMS group is being slowly activated by a fluoride from the BF₃, and then coordinating to proceed via a closed transition state? It might be interesting, though not vital, to try that reaction again with an added fluoride source (perhaps catalytic amount) to see if that could be a high-yielding syn selective reaction. I see a similar concept in Entry 18 of S2, but it lacks the BF₃, and is heated, so not the same as what I am proposing. If it worked it's a neat demonstration of each diastereomer being accessible by allylsilanes.

2. Given the previous work in the area of superacids and these atranes, I also wonder what would happen if the stoichiometric BF₃ was replaced by a catalytic amount of a superacidic atrane? I'm surprised that TMS-OTf (Entry 8, S3) didn't give any product – it would be interesting to know not just the yield of the reactions in S3 but also if any reaction took place and if any other products could be identified. This might be important for other reaction scaffolds using the developed conditions – just making researchers aware of potential alternative reaction pathways when exposing these ketones to Lewis acids and allylating reagents.

Clarity and context

The article is well-written and put into historical context.

References

References are appropriate.

Your expertise

I am not able to critically evaluate the computational work as it is outside of my expertise. The rest of the work I am fully competent to assess.

Signed,

Marc J. Adler

Reviewer #2

(Remarks to the Author)

The manuscript by Yasuda et al. developed an anti-selective alkenylation of α -oxy ketones using the previously reported Group-14 allyl atranes (Chem Asian J. 2021, 16, 3118–3123). The authors employ computational investigations to compare the reactivity of atrane-type allyl nucleophiles with that of allylSiPh₃, suggesting that the former exhibits high nucleophilicity and low chelating ability. Despite the new approach to anti-selective allylation via a non-chelation pathway, the study is not acceptable for publication due to fundamental issues that remain unclear. Firstly, the reported second-order perturbation energy ($\sigma(\text{Sn}-\text{C}\alpha) \rightarrow \pi^*(\text{C}\beta=\text{C}\gamma)$) is 9.37 kcal/mol for Sn(allyl) versus 5.61 kcal/mol for allylSiPh₃. However, a higher value indicates a stronger interaction, which is irreconcilable with the claim that transannular interactions stabilize the allyl-anion character. This contradiction invalidates the design of nucleophiles. More importantly, the origin of the anti-selectivity remains obscure. This is highlighted by the observation in Scheme 1b that both nucleophiles (allylSiMe₃ and Si(allyl)) yield the anti-selective product with cyclobutanone, suggesting the stereochemical outcome is not uniquely dictated by the nucleophile's properties as claimed. The steric effect can be a pivotal factor that affects the anti-selectivity. Consequently, without deeper mechanistic investigations, this work remains descriptive and fails to provide a coherent or convincing explanation for the observed reactivity and selectivity. In summary, due to the unresolved contradictions in the computational results and the lack of a clear mechanistic foundation, this reviewer cannot recommend the manuscript for publication in Nature Communications.

Reviewer #3

(Remarks to the Author)

Reviewer #4

(Remarks to the Author)

In the submitted manuscript, Yasuda and Konishi describe a strategy to selectively access the anti-diastereomer in the allylation of α -oxy ketones. This method addresses a difficult challenge in stereocontrolled additions to α -oxy ketones, whose chemistry is dominated by syn-diastereomer formation arising from chelation-controlled addition pathways. In this work, the authors expand upon their prior studies of Group 14 metallatrans derivatives, employing an allylcarbasilatrans derivative [1Si(allyl)] whose backbone includes aryl connectors between the metal center and the caged backbone. The authors demonstrate that a high degree of diastereocontrol can be achieved for the allylation of α -oxy ketones, using their carbasilatrans-derived nucleophile 1Si(allyl) to access the anti-diastereomer via a non-chelation-controlled mechanism and allylstannane/SnCl₂ to access the syn-diastereomer via a chelation-controlled mechanism. A key enabling factor involved in the selectivity achieved is the transannular nitrogen coordination to silicon, which results in the diminished Lewis acidity of Si while heightening its nucleophilicity relative to non-atrans-based allylsilanes, thus allowing access to a non-chelation-controlled allylation pathway. This process successfully generates the anti-diastereomer in the allylation of α -oxy aryl ketones with very high diastereoselectivity, though the use of α -oxy aldehydes results in poor selectivity, and the use of an α -oxy aliphatic ketone results in no product. Despite these substrate limitations, this is a nice demonstration of the unique reactivity that can be achieved through the variation of Group 14 metal and the atrans backbone. It not only provides a practical strategy to access anti-diastereomers from the allylation of α -oxy aldehydes, but also illustrates how the low Lewis acidity and tunable nucleophilicity of Group 14 metallatrans can allow access novel reaction pathways. I support this manuscript for publication in Nature Communication after my comments below are addressed.

The substrates in Table 2 primarily include α -aryl- α -oxy ketones. With the exception of the cyclic substrate 2j (entry 9), α -alkyl- α -oxy ketones are only used in examples exploring specialized substrates (bulky 2e, O–N-containing 2f) or in examples that fail (2k and 2l). I would like to see the inclusion of entries using an α -alkyl- α -acetoxy ketone and an α -alkyl- α -alkoxy ketone, which would make any potential limitations that might arise from the presence of an α -alkyl substituent more transparent.

The authors suggest that nucleophilicity of 1Si(allyl) is insufficient to react with the aliphatic ketone. This presents a fairly substantial limitation to the described method. Does the use of the more reactive 1Sn(allyl) result in any observed allylation? The authors should comment on the reactivity observed using 1Sn(allyl) with α -oxy aliphatic ketones.

According to the NBO data in Table S6, the C(γ) nucleophilicity of allylsilatrans is almost identical to that of 1Si(allyl). This suggests that the lack of stability of allylsilatrans in the presence of BF₃ is the sole reason why it is ineffective in this chemistry. This highlights the unique properties of 1Si(allyl) with respect to both stability and reactivity. This might be useful to mention in the text of the manuscript as it establishes a calculation-based comparison of reactivity between allylsilatrans and 1Si(allyl) that implies similar activating abilities. This is also noteworthy as allylcarbasilatrans would be expected to be significantly more nucleophilic than allylsilatrans, implying that the dampened electronic effect achieved by replacing the C(sp³)-substituents in allylcarbasilatrans with the C(sp²)-substituents of 1Si(allyl) closely parallels the effect achieved from the O substituents in allylsilatrans.

Other comments:

Page 3 (bottom): I recommend that “which exhibits high nucleophilicity” be changed to “which exhibits heightened nucleophilicity” since this is in comparison to the allylsilane.

Page 4 (top): The line “The use of 6 in the presence of SnCl₂ and allylindium provided product syn-3a in high yield with excellent syn-selectivity...” makes it sound like 6 was treated with allylindium, which is not the case according to Table 1, entry 5. This line should be re-written, more clearly indicating that allylindium alone was employed as the nucleophile in the absence of 6 and SnCl₂.

Table S6: In the left column under NBO charge, q(Si) should be q(E) as Ge and Sn values are also given.

Some NMR spectra are scaled so small that they are almost unreadable (see S35, S45, S47, 48 for example). This also creates a problem in assessing impurities in the spectra. These NMR spectra definitely need to be expanded. Any prominent impurity should also be indicated and identified.

Reviewer #5

(Remarks to the Author)

Yasuda and coworkers reported non-chelation control of alpha-oxy ketones by allylation using allylsilatrans and related group 14 allylatrans. As pointed out by the authors, the syn-selective additions of α -oxy carbonyls via a chelation pathway are well established, anti-selective additions via a non-chelation pathway remain largely unexplored. General methods to obtain anti-selective additions have not been well established. In order to overcome this inherent issue, the authors employed atrans-type allyl nucleophiles that feature a group-14-element center which have the exceptional nucleophilicity to allylation reaction of the carbonyl compounds. SI is also carefully prepared.

The conceptual background stems from using the authors' allylsilatrans as the specific allylation reagent, therefore this reviewer recommends publication of this manuscript in Nature Communications. However, the following issue to be addressed before publication.

1) The authors stated that a diverse array of α -oxy ketones were successfully carried out. In fact, a variety of protective groups of the alpha-oxy unit were applicable to this anti-selective allylation. However, the anti-selective allylation was achieved only in aryl ketone derivatives. Aliphatic ketone and aldehyde were not suited for this allylsilatrane approach. Therefore, it must be said that the present method can only be applied to a limited scope of substrates, this reviewer feels somewhat doubtful about the authors' statement.

2) More mechanistic support is needed to strengthen this manuscript, for example the transition state analysis of this particular reaction. Why is this particular anti-selective allylation achieved by using specific allylsilatrane. It is readily understandable that allylsilatrane derivatives have the high nucleophilicity without theoretical calculations, because of the hyper valent bond formation around silicon center. Current interest is "anti-selective" allylation, not high nucleophilicity of allylsilatrane (of course, it is important to achieve anti-selectivity).

3) The authors proposed brief reaction mechanism based on NMR experiments. These analyses are solid evidence for the proposed mechanism. The authors postulated that one of the fluorine atoms on the boron atom is captured by silyl cation 1Si^+ , which results in the formation of the allyl product 1nt and 1SiF . It may be possible to form 1Si^+ species because of Si having an atrane unit. However, in the allylation of cyclic ketones shown in Scheme 1a, if the allylation proceed via an open chain model, the reaction of 2o may not give the product (but high trans-selectivity was achieved). Therefore, cyclic transition state, where weak interaction between Si atom and F atom of BF_3 at the C-C-bond formation step, is also feasible.

4) The introduction of this manuscript focused on the anti-selective product formation based on the Felkin-Anh model. Therefore, the reaction of cyclic ketones is confusing. Steric bulkiness of the allylsilatrane is the major interest in these reactions. As mentioned in the above comment, this reviewer suggests that the authors reconsider how to use these experimental results.

Version 2:

Reviewer comments:

Reviewer #1

(Remarks to the Author)

The authors have sufficiently addressed all of my comments and questions.

Reviewer #2

(Remarks to the Author)

I have reviewed the authors' response and the revised manuscript. While the use of second-order perturbation energy to explain nucleophilicity differences among allylic species of the same element (Table S6 in Response to Referees Letter) is reasonable, my central concern, as noted by another reviewer, regarding the unconvincing explanation for reaction selectivity remains unresolved.

Specifically, in Table 1, non-cage allylic Si/Sn compounds yield syn-type chelation products, whereas cage-type analogues afford anti-type non-chelation products. This product distribution cannot be rationalized solely by nucleophilicity, as the second-order perturbation energies (9.37 kcal/mol for $1\text{Sn}(\text{allyl})$ vs. 5.61 kcal/mol for allylSiPh_3) do not align with the observed stereochemical outcomes. The reactivity shown in Scheme 1b further confirms that product selectivity is not governed exclusively by nucleophilicity.

Although the Group 14 metal-containing heteroatom cage derivatives developed here address the challenges associated with the anti-selective addition of α -oxy carbonyl compounds, elucidating the origin of anti-selective addition is essential to validate this new strategy. This reviewer strongly recommend that the authors provide conclusive computational evidence to explain the syn/anti-type chelation/non-chelation selectivity in the next revision.

Reviewer #3

(Remarks to the Author)

Reviewer #4

(Remarks to the Author)

I am satisfied with the additions and corrections made by the authors in response to the bulk of my comments. However, I feel that the NMR spectra still have major deficiencies. When I initially commented "Some NMR spectra are scaled so small that they are almost unreadable (see S35, S45, S47, 48 for example). This also creates a problem in assessing impurities in the spectra. These NMR spectra definitely need to be expanded. Any prominent impurity should also be indicated and identified," I did not mean to imply that these were the only problematic NMR spectra – the problem is rampant throughout

the SI. I expected that the authors would critically reassess all spectra and expand the scale of others where obviously necessary. Compare the scale of the ^1H NMR spectra on S26, S27, S58, and S61 to the spectra on S29 and S31, for example. The scale for S29 and S31 is reasonable, but why are S26, S27, S58, and S61 scaled so small? Additionally, although the authors have expanded the vertical scale for the NMR spectra on S35, S45, S47, and S48, and have labeled the impurities with asterisks, the expanded spectra show levels of impurities that I would consider inappropriate for publication. If the authors could specifically identify the impurities as a solvent or diastereomer, the spectra could potentially be acceptable, but otherwise, these spectra contain too much unidentified impurity and would need to be re-purified (S45 and S47 are particularly problematic). As I mentioned in my initial evaluation, it is impossible to assess the purity of some of these compounds with their NMR spectra scaled so small. I suspect that several other compounds also contain impurity levels that exceed what is generally acceptable in a synthetic study.

Version 3:

Reviewer comments:

Reviewer #2

(Remarks to the Author)

In this third round of revision, the authors still have not addressed the fundamental mechanistic question of the anti-selectivity origin in the allylation of α -oxy ketones. Although the authors reiterate that stereoselectivity stems from the transition state conformations of α -oxy carbonyl compounds during nucleophilic attack, the manuscript continues to lack a substantive and direct investigation into this claim. Merely citing the polar Felkin-Anh model and previous reports on α -oxy aldehydes is inadequate; it does not explain the unique stereocontrol mechanism in this new anti-selective allylation system. Since the core issue of stereoselectivity remains unresolved after two rounds of revision, the manuscript still lacks the compelling evidence and mechanistic depth required for publication in Nature Communications. I cannot recommend the publication of this work at the current stage.

Reviewer #3

(Remarks to the Author)

Reviewer #4

(Remarks to the Author)

The NMR spectra contained in the SI are significantly improved. I now find this manuscript suitable for publication in Nat. Comm.

Version 4:

Reviewer comments:

Reviewer #2

(Remarks to the Author)

In this revised version, the authors have provided DFT calculations to clarify the origin of anti-selectivity in the allylation reaction and the syn/anti selectivity of the nucleophilic attack by α -oxy carbonyl compounds. However, the Gibbs free energy data were obtained under gas-phase conditions. In real reaction systems, solvent effects can significantly influence the energy trends and alter the interpretation of selectivity. The energy obtained in the solvent based on the structures optimized in gas-phase should be used for discussion in the main text. This work can be accepted for publication after the authors update the energy data.

Reviewer #3

(Remarks to the Author)

Version 5:

Reviewer comments:

Reviewer #2

(Remarks to the Author)

The revised manuscript has now incorporated solvent effects in the DFT calculations, and the updated computational results

are consistent with the experimentally observed cis/trans selectivity in the allylation reaction. This revision now addresses this reviewer's concerns and is suitable for publication in Nature Communications.

Reviewer #3

(Remarks to the Author)

Our responses to Referee 1

Key results

This paper by Tsutsui, Konishi, and Yasuda details the selective anti-allylation of alpha-oxyketones using a bulky (and interesting) allylating agent and a monodentate Lewis acid (BF₃-OEt₃). The reaction proceeds via an open transition state, rendering the stereoselectivity adherent to the traditional Felkin-Anh model and resulting in the anti-product. This reaction is notable for the researchers identifying a solution to having to balance the nucleophilicity of the nucleophile with the Lewis acidity of the nucleophile-carrier, i.e. the atrane is not Lewis acidic at all (precluding a closed transition state, which would provide the syn product) yet the allyl group on the atrane displays enhanced nucleophilicity. The significance of this paper is multifold. On its simplest level, this reaction provides access to a structural motif that has not been readily accessible to-date. At a deeper level, it also provides a blueprint for further development of anti-selective addition reactions of alpha-oxyketones (and potentially related substrates). Finally, it further demonstrates atranes as powerful non-coordinating atom-transfer agents, an idea that has been explored previously by this group and others.

Validity

I have no concerns about the validity or robustness of the data interpretation or conclusions. I appreciate the crystal structures to provide confidence in the interpretation of the stereochemistry of the products, and extrapolation then to the NMR data (using both shifts and splitting) to characterize further products. I really enjoyed the mechanistic studies starting on page S78, highlighting the importance of order of addition in the reaction. I wish there was room to incorporate these studies into the main text.

Significance

On the one hand, the concept that one could pair a Lewis acid with a non-coordinating nucleophile to get addition to alpha-oxyketones via an open transition state and thus anti addition is not surprising. On the other hand, the Felkin-Anh model has existed for more than 50 years and this reaction hasn't previously been well-established. So the devil is in the details of what these researchers have discovered...and that devil is that these allylatranes are unique in their reactivity. So, the real significance is the continued development of atranes as versatile and tunable atom-transfer reagents. I think that is highly significant conceptually, and I believe that the authors know and believe this (given their application of the scaffold previously), though it's not a glaringly obviously important takeaway from the paper. It may be worth just adding some emphasis encouraging other synthetic organic chemists to look at atranes for this type of use...just a bit more on the potential downstream impact of these molecules up front could go a long way. Or even just sprucing up the part of the Figure 1b that highlights the unique properties of the atrane.

I also discuss some matters relating to significance below, in "Suggested Improvements".

First of all, we appreciate the positive comments and kind encouragement. According to the reviewer's suggestion, we have modified Figure 1b to highlight the unique properties of the atrane, as shown below.

Figure 1. **a** Summary for the diastereoselective addition to α -chiral carbonyls. **b** anti-Selective allylation of α -oxy ketones by 1E(allyl) (E = Si, Ge, or Sn; **this work**). **c** Calculated electronic properties of 1Si(allyl) (left) and allylSiPh₃ 10 (right). **d** Plausible reaction mechanism of the anti-selective allylation using 1Si(allyl).

The experimental work presented certainly supports the claims of the paper. There is the question of how the synthesized products might directly be useful, to which there is really no answer provided. Could one find evidence of such scaffolds in, for example, a bioactive molecule? A chiral ligand? I think it would strengthen the case for significance to see some of these products being transformed into useful molecules – and if that's not easily doable (since these motifs have been difficult to make) then perhaps one could target stereoisomers of important compounds that are known? I think it would help non-synthetic-organic-chemistry readers (who don't know or care about novel exploitation of Felkin-Anh) to understand why this is important.

The issue raised by the reviewer is scientifically significant. While we have extensively screened molecules with scaffolds accessible through our established method, we were unable to find any reports on their applications to bioactive molecules. However, it is well established that *anti*-1,2-diols serve as crucial scaffolds for chiral bidentate ligands. To address this point, we have added the following sentence to the summary part of the manuscript.

[before]

Furthermore, our allylic reagents can be expected to find broad applications as effective building blocks in natural-product synthesis.

[after]

Furthermore, it is feasible to expect that the obtained *anti*-1,2-diols will find broad applications as effective building blocks in chiral ligands.

*1. I wonder why the TMS-allyl group give *syn* selectivity? The low nucleophilicity results in low yield (as mentioned), but doesn't explain the observed diastereoselectivity. Is it possible that the TMS group is being slowly activated by a fluoride from the BF₃, and then coordinating to proceed via a closed transition state? It might be interesting, though not vital, to try that reaction again with an added fluoride source (perhaps catalytic amount) to see if that could be a high-yielding *syn* selective reaction. I see a similar concept in Entry 18 of S2, but it lacks the BF₃, and is heated, so not the same as what I am proposing. If it worked it's a neat demonstration of each diastereomer being accessible by allylsilanes.*

We appreciate the thoughtful suggestion. As the reviewer indicated, it is possible that fluoride from BF₃ could activate the silicon center, leading to the formation of a high-coordinated species that may proceed through a closed transition state. Although the allylation using allylSiMe₃ and BF₃·OEt₂ has been studied extensively, there is currently no evidence or literature reports supporting this specific mechanism. The generation of a silylium cation by the activation of a Si center using a strong Lewis acid is another possible mechanism to proceed through a closed transition state. At this point, we do not have a clear explanation for the generation of the *syn*-product. Due to the poor

electrophilicity of the α -oxy ketone, the allylation using allylSiMe₃ and ⁿBu₄NF at room temperature is unlikely to proceed.

To demonstrate the possibility of a closed transition state, we have added the following sentence to the main text.

As a model reaction, we performed the allylation of benzoin methyl ether **2a** (Table 1). When allylSiMe₃ **4** was used in the presence of BF₃·Et₂O, *syn*-product **3a** was obtained. The *syn*-selectivity may be due to either the generation of a high-coordinated or a cationic Si species³⁶ under the applied reaction conditions. The low yield (6%) of this reaction is caused by the low nucleophilicity of **4** (Table 1, entry 1).

36. Kim, K.-C. *et al.* Crystallographic Evidence for a Free Silylium Ion. *Science* **297**, 825–827 (2002).

2. Given the previous work in the area of superacids and these atranes, I also wonder what would happen if the stoichiometric BF₃ was replaced by a catalytic amount of a superacidic atrane? I'm surprised that TMS-OTf (Entry 8, S3) didn't give any product – it would be interesting to know not just the yield of the reactions in S3 but also if any reaction took place and if any other products could be identified. This might be important for other reaction scaffolds using the developed conditions – just making researchers aware of potential alternative reaction pathways when exposing these ketones to Lewis acids and allylating reagents.

We appreciate the thoughtful suggestion.

We attempted the allylation of **2a** using **1Si(allyl)** along with our atrane-type Si-cation as an activator, instead of using BF₃·OEt₂. However, no reaction occurred, and only the recovery of **2a** was observed.

In Table S3, we confirmed, by means of NMR measurements of the reaction mixture, that for all entries where no reaction occurred, substrate **2a** was recovered quantitatively, which excludes the potential formation of any by-products.

For the reaction using Me₃SiOTf, we observed rapid decomposition of **1Si(allyl)** and the recovery of **2a**, as determined by an NMR analysis. This rapid decomposition of the allyl reagent in **1Si(allyl)** likely hindered the desired allylation process.

To address these observations, we have added the corresponding results to Table S3, as outlined below.

Table S3. Screening activator and solvent in allylation of **2a** with **1Si(allyl)**.
entry	Lewis acid	solvent	yield/%	syn/anti
1	BF ₃ ·Et ₂ O	CH ₂ Cl ₂	97%	5/95
2	TiCl ₄	CH ₂ Cl ₂	93%	28/72
3	SnCl ₄	CH ₂ Cl ₂	96%	25/75
4	AlCl ₃	CH ₂ Cl ₂	85%	98/2
5	InCl ₃ /Me ₃ SiCl ^[a]	CH ₂ Cl ₂	80%	98/2
6	B(C ₆ F ₅) ₃	CH ₂ Cl ₂	0% ^[c]	–
7	Sc(OTf) ₃	CH ₂ Cl ₂	0% ^[c]	–
8	Me ₃ SiOTf	CH ₂ Cl ₂	0% ^[c, d]	–
9	BF ₃ ·Et ₂ O ^[b]	CH ₂ Cl ₂	11%	10/90
10	 [1Si][ClO ₄]	CH ₂ Cl ₂	0% ^[c]	–
11	none	CH ₂ Cl ₂	0% ^[c]	–
12	BF ₃ ·Et ₂ O	CHCl ₃	78%	13/87
13	BF ₃ ·Et ₂ O	1,2-DCE	92%	6/94
14	BF ₃ ·Et ₂ O	Et ₂ O	14%	>99/1
15	BF ₃ ·Et ₂ O	CH ₃ CN	24%	13/87
16	BF ₃ ·Et ₂ O	hexane	32%	31/69
17	BF ₃ ·Et ₂ O	toluene	87%	88/12
18	BF ₃ ·Et ₂ O	THF	0% ^[c]	–
19	BF ₃ ·Et ₂ O	DMF	0% ^[c]	–

[a] 5 mol%, [b] 10 mol%, [c] Quantitative recovery of **2a** and the absence of any by-products were verified by means of NMR measurements of the reaction mixture. [d] An NMR analysis showed the rapid decomposition of **1Si(allyl)**.

Our responses to Referee 2

The manuscript by Yasuda et al. developed an anti-selective alkenylation of α -oxy ketones using the previously reported Group-14 allyltranes (Chem Asian J. 2021, 16, 3118–3123). The authors employ computational investigations to compare the reactivity of atrane-type allyl nucleophiles with that of allylSiPh₃, suggesting that the former exhibits high nucleophilicity and low chelating ability. Despite the new approach to anti-selective allylation via a non-chelation pathway, the study is not acceptable for publication due to fundamental issues that remain unclear. Firstly, the reported second-order perturbation energy ($\sigma(\text{Sn}-\text{C}\alpha) \rightarrow \pi^*(\text{C}\beta=\text{C}\gamma)$) is 9.37 kcal/mol for Sn(allyl) versus 5.61 kcal/mol for allylSiPh₃. However, a higher value indicates a stronger interaction, which is irreconcilable with the claim that transannular interactions stabilize the allyl-anion character. This contradiction invalidates the design of nucleophiles.

We are concerned that the reviewers may have misunderstood our findings. The extent of hyperconjugation, denoted as $\sigma(\text{E}-\text{C}\alpha) \rightarrow \pi^*(\text{C}=\text{C})$, is more pronounced when the atomic number of element E increases. Consequently, when comparing homologs that contain identical substituents or ligands, the second-order perturbation energy tends to increase in the order of Si < Ge < Sn. However, the reviewer's point is that the comparison is between totally different allylic species with different frameworks. Thus, the pointed comparison is not suitable for discussion.

We have calculated the value for allylSnPh₃ to be 9.53 kcal/mol, which is higher than that for 1Sn(allyl) (9.37 kcal/mol). This estimation also supports our arguments. We have added the calculated values for allylSnPh₃ to Table S6, Figure S14, and Figure S15.

Table S6. Summary of parameters related to nucleophilicity.

	1Si(allyl)	1Ge(allyl)	1Sn(allyl)	allylSiPh ₃	allylSi(OC ₂ H ₄) ₃ N	allylSnPh ₃
HOMO /eV	 -6.18	 -6.02	 -5.91	 -6.60	 -6.17	 -6.37
NBO charge						
q(E)	+1.891	+1.697	+1.898	+1.872	+2.379	+1.852
q(C α)	-1.038	-0.970	-0.986	-1.009	-1.061	-0.958
q(C β)	-0.215	-0.222	-0.230	-0.227	-0.217	-0.244
q(C γ)	-0.492	-0.491	-0.506	-0.481	-0.491	-0.491
$\sigma(\text{E}-\text{C}\alpha) \rightarrow \pi^*(\text{C}\beta=\text{C}\gamma)$ /kcal/mol	3.84	5.29	9.37	5.61	5.90	9.53
$\Delta\delta_{\text{C}\beta-\text{C}\gamma}$ /ppm	25.3	24.4	28.4	18.7	27.8	23.5

B3PW91/DGDZVP (for Si, Ge, and Sn), 6-31+G**(for C, H, and N) level.

Figure S14. HOMO of (a) 1Si(allyl), (b) 1Ge(allyl), (c) 1Sn(allyl), (d) allylSiPh₃, (e) allylSi(OC₂H₄)₃N, and (f) allylSnPh₃ calculated at the B3PW91/DGDZVP (for Si, Ge, and Sn), 6-31+G**(for C, H, and N) level.

Figure S15. Natural bond orbitals of (a) 1Si(allyl), (b) 1Ge(allyl), (c) 1Sn(allyl), (d) allylSiPh₃, (e) allylSi(OC₂H₄)₃N, and (f) allylSnPh₃ calculated at the B3PW91/DGDZVP (for Si, Ge, and Sn), 6-31+G**(for C, H, and N) level.

More importantly, the origin of the anti-selectivity remains obscure. This is highlighted by the observation in Scheme 1b that both nucleophiles (allylSiMe₃ and Si(allyl)) yield the anti-selective product with cyclobutanone, suggesting the stereochemical outcome is not uniquely dictated by the nucleophile's properties as claimed. The steric effect can be a pivotal factor that affects the anti-selectivity. Consequently, without deeper mechanistic investigations, this work remains descriptive and fails to provide a coherent or convincing explanation for the observed reactivity and selectivity.

The issue that the reviewer pointed out is scientifically significant. We are very sorry for any potentially unclear discussions on this point, because the mechanism for the stereoselectivity in the allylation of cyclic ketones is not directly explained based on the Felkin-Anh model. In these reactions, the main interest is the steric demand of the allylsilatrane. To clarify this fact, we have modified the corresponding sentence in Scheme 1b, as shown below.

Furthermore, the steric effect of **1**Si(allyl) was evident in the allylation of α -oxy cyclobutanone **2q** (Scheme 1b). The resulting allylated product (**3q**) is an important synthetic intermediate for the substructure of a β -secretase modulator candidate.

Our responses to Referee 3

We appreciate the reviewer's dedication.

Our responses to Referee 4

In the submitted manuscript, Yasuda and Konishi describe a strategy to selectively access the anti-diastereomer in the allylation of α -oxy ketones. This method addresses a difficult challenge in stereocontrolled additions to α -oxy ketones, whose chemistry is dominated by syn-diastereomer formation arising from chelation-controlled addition pathways. In this work, the authors expand upon their prior studies of Group 14 metallatrane derivatives, employing an allylcarbasilatrane derivative [1Si(allyl)] whose backbone includes aryl connectors between the metal center and the caged backbone. The authors demonstrate that a high degree of diastereocontrol can be achieved for the allylation of α -oxy ketones, using their carbasilatrane-derived nucleophile 1Si(allyl) to access the anti-diastereomer via a non-chelation-controlled mechanism and allylstannane/SnCl₂ to access the syn-diastereomer via a chelation-controlled mechanism. A key enabling factor involved in the selectivity achieved is the transannular nitrogen coordination to silicon, which results in the diminished Lewis acidity of Si while heightening its nucleophilicity relative to non-atrane-based allylsilanes, thus allowing access to a non-chelation-controlled allylation pathway. This process successfully generates the anti-diastereomer in the allylation of α -oxy aryl ketones with very high diastereoselectivity, though the use of α -oxy aldehydes results in poor selectivity, and the use of an α -oxy aliphatic ketone results in no product. Despite these substrate limitations, this is a nice demonstration of the unique reactivity that can be achieved through the variation of Group 14 metal and the atrane backbone. It not only provides a practical strategy to access anti-diastereomers from the allylation of α -oxy aldehydes, but also illustrates how the low Lewis acidity and tunable nucleophilicity of Group 14 metallatranes can allow access novel reaction pathways. I support this manuscript for publication in Nature Communication after my comments below are addressed.

We really appreciate the very positive comments.

The substrates in Table 2 primarily include α -aryl- α -oxy ketones. With the exception of the cyclic substrate 2j (entry 9), α -alkyl- α -oxy ketones are only used in examples exploring specialized substrates (bulky 2e, O–N-containing 2f) or in examples that fail (2k and 2l). I would like to see the inclusion of entries using an α -alkyl- α -acetoxy ketone and an α -alkyl- α -alkoxy ketone, which would make any potential limitations that might arise from the presence of an α -alkyl substituent more transparent.

We appreciate the kind suggestions. Although we have not attempted the allylations of α -alkyl- α -acetoxy ketones, the allylation of α -methyl- α -methoxy ketone **2j** using 1Si(allyl) and BF₃·OEt₂ has been demonstrated. The product **3j** was obtained in 97% yield with a high anti-selectivity (syn/anti = 4/96). The result has been added to entry 9 in Table 2. We have renumbered the compound numbers after adding α -methyl- α -methoxy ketone **2j** to Table 2.

The reaction also gave good yields and diastereoselectivities with *p*-chloro- or *p*-phenyl-substituted benzoin methyl ethers (**2g**, **2h**) and a naphthyl ketone derivative (**2i**) (conditions A, Table 2, entries 6-8). Furthermore, the allylations of α -methyl- α -methoxy ketones **2j** and **2k** with a 2,5-disubstituted *trans*-tetrahydrofuran motif afforded **3j** and **3k** with nearly perfect *anti*-selectivity (*syn/anti* = 4/96 and 2/98) (conditions A, Table 2, entries 9–10). The selectivity of **3k** was dramatically improved relative to a previous study with allylMgCl (*syn/anti* = 29/71).⁴⁶

Entry 9 in Table 2.

The authors suggest that nucleophilicity of 1Si(allyl) is insufficient to react with the aliphatic ketone. This presents a fairly substantial limitation to the described method. Does the use of the more reactive 1Sn(allyl) result in any observed allylation? The authors should comment on the reactivity observed using 1Sn(allyl) with α -oxy aliphatic ketones.

We appreciate the kind suggestions. We have attempted the allylation of **2k** (now renamed **2l**) using **1Sn(allyl)** with $\text{BF}_3 \cdot \text{OEt}_2$, but no reaction occurred and the substrate was recovered quantitatively. Accordingly, we have added the following sentence to the main text.

Unfortunately, the desired product was not obtained from the allylation of α -oxy aliphatic ketone **2l** (conditions A, Table 2, entry 11), as the allylation of aliphatic ketones requires an allyl nucleophile with higher nucleophilicity; using **1Sn(allyl)** instead of **1Si(allyl)** had no effect.

According to the NBO data in Table S6, the $C(\text{gamma})$ nucleophilicity of allylsilatrane is almost identical to that of 1Si(allyl). This suggests that the lack of stability of allylsilatrane in the presence of BF_3 is the sole reason why it is ineffective in this chemistry. This highlights the unique properties of 1Si(allyl) with respect to both stability and reactivity. This might be useful to mention in the text of the manuscript as it establishes a calculation-based comparison of reactivity between allylsilatrane and 1Si(allyl) that implies similar activating abilities. This is also noteworthy as allylcarbasilatrane would be expected to be significantly more nucleophilic than allylsilatrane, implying that the dampened electronic effect achieved by replacing the $C(\text{sp}^3)$ -substituents in allylcarbasilatrane with the $C(\text{sp}^2)$ -substituents of 1Si(allyl) closely parallels the effect achieved from the O substituents in allylsilatrane.

We appreciate the kind suggestions. According to the reviewer's suggestion, we have added the following explanations to the main text in order to highlight the unique properties of **1Si(allyl)**.

To investigate the electronic properties of **1E(allyl)**, we calculated its molecular orbitals (Figure 1c). The

calculated energy of the HOMO of **1Si(allyl)** (-6.18 eV) suggests that its nucleophilicity is higher than that of allylSiPh₃ **10** (-6.60 eV). The high nucleophilicity of **1Si(allyl)** is also supported by ¹³C NMR and natural-bond-orbital (NBO) analyses (Table S6). The difference in NBO charge between C_β and C_γ was higher for **1Si(allyl)** (|q(C_β)-q(C_γ)| = 0.277) than for **10** (|q(C_β)-q(C_γ)| = 0.254). Meanwhile, a second-order-perturbation analysis indicated that the hyperconjugation from the σ(Si-C_α) bond to the π*(C_β=C_γ) bond is less effective in **1Si(allyl)** (ΔE = 3.84 kcal/mol) than in **10** (ΔE = 5.61 kcal/mol), commensurate with weak electronic communication between the allylic moiety and the group-14-element center of **1Si(allyl)**. Thus, the strong nucleophilic character of **1Si(allyl)** should reflect the charge localization rather than the stereoelectronic effects. Notably, the results of the NBO analyses indicate that the nucleophilicity of **1Si(allyl)** is similar to that of allylsilatrane **5**, which rapidly decomposes under the applied reaction conditions. This highlights the balance of reactivity and stability in **1Si(allyl)**, facilitated by its rigid triaryl-based structure. The stabilization of the silyl center by transannular interactions would also enhance the allyl-anion character. A crystallographic analysis confirmed that the molecular geometry of **1Si(allyl)** exhibits a nearly trigonal-bipyramidal structure with a five-coordinated silicon center.

Page 3 (bottom): I recommend that “which exhibits high nucleophilicity” be changed to “which exhibits heightened nucleophilicity” since this is in comparison to the allylsilane.

We appreciate the kind suggestion. We have changed the part in accordance with the reviewer's suggestion.

Oxygen-ligated allylsilatrane allylSi(OC₂H₄)₃N **5**, which exhibits heightened nucleophilicity,^{37,38} was an ineffective allylation reagent (Table 1, entry 2). Nuclear-magnetic-resonance (NMR) measurements revealed that **5** decomposes rapidly in the presence of BF₃·Et₂O (Figure S10).

Page 4 (top): The line “The use of 6 in the presence of SnCl2 and allylindium provided product syn-3a in high yield with excellent syn-selectivity...” makes it sound like 6 was treated with allylindium, which is not the case according to Table 1, entry 5. This line should be re-written, more clearly indicating that allylindium alone was employed as the nucleophile in the absence of 6 and SnCl2.

We are very sorry for the unclear description. We have changed the part, as shown below.

Here, stannyl cation species generated by transmetalation with BF₃·Et₂O likely activate the substrate in a chelation-controlled manner.^{39,40} Treating **2a** with allylSnBu₃/SnCl₂⁴¹ or allylindium^{23,24,42} provided product **syn-3a** in high yield with excellent *syn*-selectivity (*syn/anti* = >99/1; Table 1, entries 4 and 5) via the chelation pathway. The reactions with allylMgBr **8** or allylLi **9** proceeded without diastereoselectivity due to strong chelation combined with too high nucleophilicity (Table 1, entries 6 and 7).^{43,44}

Table S6: In the left column under NBO charge, $q(\text{Si})$ should be $q(\text{E})$ as Ge and Sn values are also given.

We appreciate the reviewer's findings. We have corrected these parts.

Table S6. Summary of parameters related to nucleophilicity.

	1Si(allyl)	1Ge(allyl)	1Sn(allyl)	allylSiPh ₃	allylSi(OC ₂ H ₄) ₃ N	allylSnPh ₃
HOMO /eV	 -6.18	 -6.02	 -5.91	 -6.60	 -6.17	 -6.37
NBO charge						
$q(\text{E})$	+1.891	+1.697	+1.898	+1.872	+2.379	+1.852
$q(\text{C}_\alpha)$	-1.038	-0.970	-0.986	-1.009	-1.061	-0.958
$q(\text{C}_\beta)$	-0.215	-0.222	-0.230	-0.227	-0.217	-0.244
$q(\text{C}_\gamma)$	-0.492	-0.491	-0.506	-0.481	-0.491	-0.491
$\sigma(\text{E}-\text{C}_\alpha) \rightarrow \pi^*(\text{C}_\beta-\text{C}_\gamma)$ /kcal/mol	3.84	5.29	9.37	5.61	5.90	9.53
$\Delta\delta_{\text{C}_\beta-\text{C}_\gamma}$ /ppm	25.3	24.4	28.4	18.7	27.8	23.5

B3PW91/DGDZVP (for Si, Ge, and Sn), 6-31+G**(for C, H, and N) level.

Some NMR spectra are scaled so small that they are almost unreadable (see S35, S45, S47, 48 for example). This also creates a problem in assessing impurities in the spectra. These NMR spectra definitely need to be expanded. Any prominent impurity should also be indicated and identified.

We are very sorry for these shortcomings. We have added expanded figures with some notations for impurities to the SI.

pS35

¹H NMR: (400 MHz, CDCl₃)

Asterisks represent inseparable impurities and residual solvents.

pS45

¹H NMR: (400 MHz, CDCl₃)

Asterisks represent inseparable impurities and residual solvents.

pS47

¹H NMR: (400 MHz, benzene-d₆)

Asterisks represent inseparable impurities and residual solvents.

pS48

¹H NMR: (400 MHz, CDCl₃)

Asterisks represent inseparable impurities and residual solvents.

Our responses to Referee 5

Yasuda and coworkers reported non-chelation control of alpha-oxy ketones by allylation using allylsilatrane and related group 14 allylitrane. As pointed out by the authors, the syn-selective additions of α -oxy carbonyls via a chelation pathway are well established, anti-selective additions via a non-chelation pathway remain largely unexplored. General methods to obtain anti-selective additions have not been well established. In order to overcome this inherent issue, the authors employed atrane-type allyl nucleophiles that feature a group-14-element center which have the exceptional nucleophilicity to allylation reaction of the carbonyl compounds. SI is also carefully prepared.

The conceptual background stems from using the authors' allylsilatrane as the specific allylation reagent, therefore this reviewer recommends publication of this manuscript in Nature Communications. However, the following issue to be addressed before publication.

We really appreciate the very positive comments.

1) The authors stated that a diverse array of α -oxy ketones were successfully carried out. In fact, a variety of protective groups of the alpha-oxy unit were applicable to this anti-selective allylation. However, the anti-selective allylation was achieved only in aryl ketone derivatives. Aliphatic ketone and aldehyde were not suited for this allylsilatrane approach. Therefore, it must be said that the present method can only be applied to a limited scope of substrates, this reviewer feels somewhat doubtful about the authors' statement.

We agree with the issue that the reviewer pointed out. We have changed the corresponding part as shown below.

Through comprehensive experimental and theoretical investigations, we demonstrate that the exceptional nucleophilicity of an allylsilatrane is primarily driven by charge localization rather than traditional stereoelectronic effects. Anti-selective allylations of **various α -aryl- α -oxy ketones** were successfully carried out. Our results effectively address long-standing challenges associated with the anti-selective addition of α -oxy carbonyl compounds.

2) More mechanistic support is needed to strengthen this manuscript, for example the transition state analysis of this particular reaction. Why is this particular anti-selective allylation achieved by using specific allylsilatrane. It is readily understandable that allylsilatrane derivatives have the high nucleophilicity without theoretical calculations, because of the hyper valent bond formation around silicon center. Current interest is “anti-selective” allylation, not high nucleophilicity of allylsilatrane (of course, it is important to achieve anti-selectivity).

We appreciate the reviewer's suggestions and recognize the significance of theoretical calculations regarding the reaction pathway. We have continued our theoretical calculations even after the submission of our work. However, progress has been challenging due to the necessity of performing numerous conformational analyses in the polar Felkin-Anh model, which involves a non-cyclic transition state. As a result, we currently do not have deeper theoretical insights into the reaction pathway. We have initiated collaborations with a theoretical chemist to investigate the reaction mechanism that leads to the observed selectivity. To clarify this situation, we have added the following sentence to the main text.

Therefore, it is considered that this *anti*-selective allylation proceeds via a non-chelation pathway. Further theoretical calculations to investigate the pathway are currently in progress in our group. The high stability of 1Si(allyl) derived from the atrane structure suppresses the undesired *in-situ* exchange process.

3) The authors proposed brief reaction mechanism based on NMR experiments. These analyses are solid evidence for the proposed mechanism. The authors postulated that one of the fluorine atoms on the boron atom is captured by silyl cation 1Si⁺, which results in the formation of the allyl product Int and 1SiF. It may be possible to form 1Si⁺ species because of Si having an atrane unit. However, in the allylation of cyclic ketones shown in Scheme 1a, if the allylation proceed via an open chain model, the reaction of 2o may not give the product (but high *trans*-selectivity was achieved). Therefore, cyclic transition state, where weak interaction between Si atom and F atom of BF₃ at the C-C-bond formation step, is also feasible.

We apologize for not clarifying that substrate **2o** (now renamed **2p**) is an unusual α -oxy-carbonyl. The product (**3p**) resulting from the allylation of **2p** exhibits the same stereo-configuration (*trans*) through both the chelation and non-chelation pathways. In the chelation pathway, a twisted-boat transition-state structure has been proposed, which leads to the *trans* product (L. A. Paquette *et al.*, *JOC* **1998**, 5604). Conversely, in our non-chelation pathway, the steric hindrance of 1Si(allyl) favors the attack on the equatorial face of **2p**, resulting in the *trans* product. To clarify the point, we have added the following explanation and a citation to the main text.

Then, the reactivity of allylsilatrane 1Si(allyl) with cyclic α -oxy ketones was explored (Scheme 1a). In the case of 2-methoxycyclohexanone **2n**, low diastereoselectivity was observed (*cis/trans* = 62/38), probably

due to the flexible conformational orientation of the methoxy group. In contrast, the allylations of conformationally fixed cyclohexanones **2o** and **2p** with a ^tBu group at the 4-position, selectively yielded diastereomers via an equatorial attack. For the allylation of **2p**, the product showed the *trans*-configuration, which is identical to that generated under the reported chelation conditions involving a twist-boat conformation in the transition state.⁴⁷ These results indicate that **1Si(allyl)** acts as a bulky allylic nucleophile, favoring nucleophilic attack from the less hindered equatorial position under our non-chelation conditions.

47. Paquette, L. A. & Lobben, P. C. Evaluation of Chelation Effects Operative during Diastereoselective Addition of the Allylindium Reagent to 2- and 3-Hydroxycyclohexanones in Aqueous, Organic, and Mixed Solvent Systems. *J. Org. Chem.* **63**, 5604–5616 (1998).

4) The introduction of this manuscript focused on the anti-selective product formation based on the Felkin-Anh model. Therefore, the reaction of cyclic ketones is confusing. Steric bulkiness of the allylsilatrane is the major interest in these reactions. As mentioned in the above comment, this reviewer suggests that the authors reconsider how to use these experimental results.

The issue that the reviewer pointed out is scientifically significant. We are very sorry for any potentially unclear discussions on this point, because the mechanism for the stereoselectivity in the allylation of cyclic ketones is not directly explained based on the Felkin-Anh model. In these reactions, the main interest is the steric bulk of the allylsilatrane. To clarify this fact, we have modified the corresponding sentence in Scheme 1b, as shown below.

Furthermore, the steric effect of **1Si(allyl)** was evident in the allylation of α -oxy cyclobutanone **2q** (Scheme 1b). The resulting allylated product (**3q**) is an important synthetic intermediate for the substructure of a β -secretase modulator candidate.

Others

Adding an author to the list of authors

Based on the reviewer's suggestion to conduct additional experiments, Mr. K. Shiga was added as a co-author.

Yuya Tsutsui,^[a] Kokoro Shiga,^[a] Akihito Konishi,^{*[a,b]} and Makoto Yasuda^{*[a,b]}

Adding some financial support to acknowledgement

We have added sources of financial support to the acknowledgments section. These were required to conduct the additional experiments.

Acknowledgements

This work was financially supported by the MEXT Grant-in-Aid for Transformative Research Areas (A) “Digitalization-driven Transformative Organic Synthesis (Digi-TOS)” (JP21H05212 [MY]), by JST CREST (JPMJCR20R3 [MY]), and by the Japan Society for the Promotion of Science (JP25K22182 [MY] and JP23H01950 [AK]). A.K. would like to express his gratitude for a “Condensed Conjugation” grant (JP23H04028 [AK]). Part of this work was supported by the JST FOREST Program (under grant number JPMJFR2426 [AK]). The authors acknowledge the Analytical Instrumentation Facility, Graduate School of Engineering, Osaka University.

Our responses to Referee 1 in the second round of revisions

The authors have sufficiently addressed all of my comments and questions.

We appreciate the reviewer's acceptance of our revisions. The reviewer's suggestions have undoubtedly improved our manuscript.

First round of the review process

Key results

This paper by Tsutsui, Konishi, and Yasuda details the selective anti-allylation of alpha-oxyketones using a bulky (and interesting) allylating agent and a monodentate Lewis acid (BF₃-OEt₂). The reaction proceeds via an open transition state, rendering the stereoselectivity adherent to the traditional Felkin-Anh model and resulting in the anti-product. This reaction is notable for the researchers identifying a solution to having to balance the nucleophilicity of the nucleophile with the Lewis acidity of the nucleophile-carrier, i.e. the atrane is not Lewis acidic at all (precluding a closed transition state, which would provide the syn product) yet the allyl group on the atrane displays enhanced nucleophilicity. The significance of this paper is multifold. On its simplest level, this reaction provides access to a structural motif that has not been readily accessible to-date. At a deeper level, it also provides a blueprint for further development of anti-selective addition reactions of alpha-oxyketones (and potentially related substrates). Finally, it further demonstrates atranes as powerful non-coordinating atom-transfer agents, an idea that has been explored previously by this group and others.

Validity

I have no concerns about the validity or robustness of the data interpretation or conclusions. I appreciate the crystal structures to provide confidence in the interpretation of the stereochemistry of the products, and extrapolation then to the NMR data (using both shifts and splitting) to characterize further products. I really enjoyed the mechanistic studies starting on page S78, highlighting the importance of order of addition in the reaction. I wish there was room to incorporate these studies into the main text.

Significance

On the one hand, the concept that one could pair a Lewis acid with a non-coordinating nucleophile to get addition to alpha-oxyketones via an open transition state and thus anti addition is not surprising. On the other hand, the Felkin-Anh model has existed for more than 50 years and this reaction hasn't previously been well-established. So the devil is in the details of what these researchers have discovered...and that devil is that these allylatranes are unique in their reactivity. So, the real significance is the continued development of atranes as versatile and tunable atom-transfer reagents. I think that is highly significant conceptually, and I believe that the authors know and believe this (given their application of the scaffold previously), though it's not a glaringly obviously important takeaway from the paper. It may be worth just adding some emphasis encouraging other synthetic organic chemists to look at atranes for this type of use...just a bit more on the potential downstream impact of these molecules up front could go a long way. Or even just sprucing up the part of the Figure 1b that highlights the unique properties of the atrane.

I also discuss some matters relating to significance below, in "Suggested Improvements".

First of all, we appreciate the positive comments and kind encouragement. According to the reviewer's suggestion, we have modified Figure 1b to highlight the unique properties of the atrane, as shown below.

Figure 1. a Summary for the diastereoselective addition to α -chiral carbonyls. b anti-Selective allylation of α -oxy ketones by 1E(allyl) (E = Si, Ge, or Sn; **this work**). c Calculated electronic properties of 1Si(allyl) (left) and allylSiPh₃ 10 (right). d Plausible reaction mechanism of the anti-selective allylation using 1Si(allyl).

The experimental work presented certainly supports the claims of the paper. There is the question of how the synthesized products might directly be useful, to which there is really no answer provided. Could one find evidence of such scaffolds in, for example, a bioactive molecule? A chiral ligand? I think it would strengthen the case for significance to see some of these products being transformed into useful molecules – and if that's not easily doable (since these motifs have been difficult to make) then perhaps one could target stereoisomers of important compounds that are known? I think it would help non-synthetic-organic-chemistry readers (who don't know or care about novel exploitation of Felkin-Anh) to understand why this is important.

The issue raised by the reviewer is scientifically significant. While we have extensively screened molecules with scaffolds accessible through our established method, we were unable to find any reports on their applications to bioactive molecules. However, it is well established that anti-1,2-diols serve as crucial scaffolds for chiral bidentate ligands. To address this point, we have added the following sentence to the summary part of the manuscript.

[before]

Furthermore, our allylic reagents can be expected to find broad applications as effective building blocks in natural-product synthesis.

[after]

Furthermore, it is feasible to expect that the obtained *anti*-1,2-diols will find broad applications as effective building blocks in chiral ligands.

1. I wonder why the TMS-allyl group give syn selectivity? The low nucleophilicity results in low yield (as mentioned), but doesn't explain the observed diastereoselectivity. Is it possible that the TMS group is being slowly activated by a fluoride from the BF₃, and then coordinating to proceed via a closed transition state? It might be interesting, though not vital, to try that reaction again with an added fluoride source (perhaps catalytic amount) to see if that could be a high-yielding syn selective reaction. I see a similar concept in Entry 18 of S2, but it lacks the BF₃, and is heated, so not the same as what I am proposing. If it worked it's a neat demonstration of each diastereomer being accessible by allylsilanes.

We appreciate the thoughtful suggestion. As the reviewer indicated, it is possible that fluoride from BF₃ could activate the silicon center, leading to the formation of a high-coordinated species that may proceed through a closed transition state. Although the allylation using allylSiMe₃ and BF₃·OEt₂ has been studied extensively, there is currently no evidence or literature reports supporting this specific mechanism. The generation of a silylium cation by the activation of a Si center using a strong Lewis acid is another possible mechanism to proceed through a closed transition state. At this point, we do not have a clear explanation for the generation of the *syn*-product. Due to the poor electrophilicity of the α -oxy ketone, the allylation using allylSiMe₃ and ⁿBu₄NF at room temperature is unlikely to proceed.

To demonstrate the possibility of a closed transition state, we have added the following sentence to the main text.

As a model reaction, we performed the allylation of benzoin methyl ether **2a** (Table 1). When allylSiMe₃ **4** was used in the presence of BF₃·Et₂O, *syn*-product **3a** was obtained. The *syn*-selectivity may be due to either the generation of a high-coordinated or a cationic Si species³⁶ under the applied reaction conditions. The low yield (6%) of this reaction is caused by the low nucleophilicity of **4** (Table 1, entry 1).

36. Kim, K.-C. *et al.* Crystallographic Evidence for a Free Silylium Ion. *Science* **297**, 825–827 (2002).

2. Given the previous work in the area of superacids and these atranes, I also wonder what would happen if the stoichiometric BF₃ was replaced by a catalytic amount of a superacidic atrane? I'm surprised that TMS-OTf (Entry 8, S3) didn't give any product – it would be interesting to know not just the yield of the reactions in S3 but also if any reaction took place and if any other products could be identified. This might be important for other reaction scaffolds using the developed conditions – just making researchers aware of potential alternative reaction pathways when exposing these ketones to Lewis acids and allylating reagents.

We appreciate the thoughtful suggestion.

We attempted the allylation of **2a** using **1Si(allyl)** along with our atrane-type Si-cation as an activator, instead of using BF₃·OEt₂. However, no reaction occurred, and only the recovery of **2a** was observed.

In Table S3, we confirmed, by means of NMR measurements of the reaction mixture, that for all entries where no reaction occurred, substrate **2a** was recovered quantitatively, which excludes the potential formation of any by-products.

For the reaction using Me₃SiOTf, we observed rapid decomposition of **1Si(allyl)** and the recovery of **2a**, as determined by an NMR analysis. This rapid decomposition of the allyl reagent in **1Si(allyl)** likely hindered the desired allylation process.

To address these observations, we have added the corresponding results to Table S3, as outlined below.

Table S3. Screening activator and solvent in allylation of **2a** with **1Si**(allyl).
entry	Lewis acid	solvent	yield/%	syn/anti
1	BF ₃ ·Et ₂ O	CH ₂ Cl ₂	97%	5/95
2	TiCl ₄	CH ₂ Cl ₂	93%	28/72
3	SnCl ₄	CH ₂ Cl ₂	96%	25/75
4	AlCl ₃	CH ₂ Cl ₂	85%	98/2
5	InCl ₃ /Me ₃ SiCl ^[a]	CH ₂ Cl ₂	80%	98/2
6	B(C ₆ F ₅) ₃	CH ₂ Cl ₂	0% ^[c]	–
7	Sc(OTf) ₃	CH ₂ Cl ₂	0% ^[c]	–
8	Me ₃ SiOTf	CH ₂ Cl ₂	0% ^[c, d]	–
9	BF ₃ ·Et ₂ O ^[b]	CH ₂ Cl ₂	11%	10/90
10	 [1Si][ClO ₄]	CH ₂ Cl ₂	0% ^[c]	–
11	none	CH ₂ Cl ₂	0% ^[c]	–
12	BF ₃ ·Et ₂ O	CHCl ₃	78%	13/87
13	BF ₃ ·Et ₂ O	1,2-DCE	92%	6/94
14	BF ₃ ·Et ₂ O	Et ₂ O	14%	>99/1
15	BF ₃ ·Et ₂ O	CH ₃ CN	24%	13/87
16	BF ₃ ·Et ₂ O	hexane	32%	31/69
17	BF ₃ ·Et ₂ O	toluene	87%	88/12
18	BF ₃ ·Et ₂ O	THF	0% ^[c]	–
19	BF ₃ ·Et ₂ O	DMF	0% ^[c]	–

[a] 5 mol%, [b] 10 mol%, [c] Quantitative recovery of **2a** and the absence of any by-products were verified by means of NMR measurements of the reaction mixture. [d] An NMR analysis showed the rapid decomposition of **1Si**(allyl).

Our responses to Referee 2 in the second round of revisions

I have reviewed the authors' response and the revised manuscript. While the use of second-order perturbation energy to explain nucleophilicity differences among allylic species of the same element (Table S6 in Response to Referees Letter) is reasonable, my central concern, as noted by another reviewer, regarding the unconvincing explanation for reaction selectivity remains unresolved.

Specifically, in Table 1, non-cage allylic Si/Sn compounds yield syn-type chelation products, whereas cage-type analogues afford anti-type non-chelation products. This product distribution cannot be rationalized solely by nucleophilicity, as the second-order perturbation energies (9.37 kcal/mol for 1Sn(allyl) vs. 5.61 kcal/mol for allylSiPh₃) do not align with the observed stereochemical outcomes. The reactivity shown in Scheme 1b further confirms that product selectivity is not governed exclusively by nucleophilicity.

Although the Group 14 metal-containing heteroatom cage derivatives developed here address the challenges associated with the anti-selective addition of α -oxy carbonyl compounds, elucidating the origin of anti-selective addition is essential to validate this new strategy. This reviewer strongly recommend that the authors provide conclusive computational evidence to explain the syn/anti-type chelation/non-chelation selectivity in the next revision.

We would like to clarify that the nucleophilicity estimated based on second-order perturbation energy is not suitable for explaining the stereoselectivity of allylations involving α -oxy ketones. In fact, we do not rely on nucleophilicity to account for the observed stereoselectivity. It is important to note here that the nucleophilicity of allylic reagents, which is derived from the electronic properties of the allyl moiety, does not have a direct connection to the stereoselectivity in their nucleophilic additions.

Instead, the stereoselectivity is determined by the transition-state conformations of α -oxy carbonyl compounds during the nucleophile attack. Based on seminal studies (for example, ref. 9), it has been empirically and theoretically accepted that α -oxy carbonyls induce *anti*-adducts in their allylations via a non-chelation pathway following the polar Felkin-Anh model. Reetz and Evans (see refs. 9, 31, and 33) have explained the *anti*-selectivity exhibited by α -oxy aldehydes due to their high electrophilicity. Given that a fundamental understanding of *syn/anti*-selectivity in nucleophilic additions to α -substituted carbonyls is well-established through abundant theoretical and experimental studies, we would like to argue that further computational evidence for our reactions is unnecessary.

In contrast to these pioneering studies, achieving *anti*-selective additions to α -oxy ketones—which are less reactive than the corresponding aldehydes as demonstrated by Reetz and Evans—remains elusive. Our manuscript addresses this critical issue in allylations. For the allylation of α -oxy ketones, it is essential to enhance either the electrophilicity of the ketone moiety or the nucleophilicity of the allylic reagent. However, both approaches encounter a significant and unavoidable challenge in achieving *anti*-selectivity. Traditionally, reagents with strong Lewis acidity are employed in these

reactions; however, they can increase both the electrophilicity of the ketone group and the nucleophilicity of the allylic reagent. Unfortunately, this often results in chelation-controlled additions that favor *syn*-selective outcomes.

To address this inherent issue and realize *anti*-selectivity in the allylation of α -oxy ketones, our goal is to harmonize high nucleophilicity with low Lewis acidity in an allylic reagent. This concept is central to our atrane-type reagent. To underscore this important point, we have added the following sentence to the main text.

that the molecular geometry of **1Si(allyl)** exhibits a nearly trigonal-bipyramidal structure with a five-coordinated silicon center. Most allylic reagents with high nucleophilicity feature a strong Lewis-acidic metal center, which facilitates the allylation of α -oxy ketones through a chelation pathway, resulting in *syn*-products.^{9,31,33} In contrast to the traditional methods for achieving *syn*-selective allylations, **1E(allyl)** enhances the nucleophilicity of the allylic group while simultaneously reducing the Lewis acidity of the metal center. This careful balance between high nucleophilicity and weak Lewis acidity in **1E(allyl)**, achieved through an atrane-type framework, allows for the distinctive *anti*-selective allylations of α -oxy ketones.

First round of the review process

The manuscript by Yasuda et al. developed an anti-selective alkenylation of α -oxy ketones using the previously reported Group-14 allyltranes (Chem Asian J. 2021, 16, 3118–3123). The authors employ computational investigations to compare the reactivity of atrane-type allyl nucleophiles with that of allylSiPh₃, suggesting that the former exhibits high nucleophilicity and low chelating ability. Despite the new approach to anti-selective allylation via a non-chelation pathway, the study is not acceptable for publication due to fundamental issues that remain unclear. Firstly, the reported second-order perturbation energy ($\sigma(\text{Sn}-\text{C}\alpha) \rightarrow \pi^*(\text{C}\beta=\text{C}\gamma)$) is 9.37 kcal/mol for Sn(allyl) versus 5.61 kcal/mol for allylSiPh₃. However, a higher value indicates a stronger interaction, which is irreconcilable with the claim that transannular interactions stabilize the allyl-anion character. This contradiction invalidates the design of nucleophiles.

We are concerned that the reviewers may have misunderstood our findings. The extent of hyperconjugation, denoted as $\sigma(\text{E}-\text{C}\alpha) \rightarrow \pi^*(\text{C}=\text{C})$, is more pronounced when the atomic number of element E increases. Consequently, when comparing homologs that contain identical substituents or ligands, the second-order perturbation energy tends to increase in the order of Si < Ge < Sn. However, the reviewer's point is that the comparison is between totally different allylic species with different frameworks. Thus, the pointed comparison is not suitable for discussion.

We have calculated the value for allylSnPh₃ to be 9.53 kcal/mol, which is higher than that for 1Sn(allyl) (9.37 kcal/mol). This estimation also supports our arguments. We have added the calculated values for allylSnPh₃ to Table S6, Figure S14, and Figure S15.

Table S6. Summary of parameters related to nucleophilicity.

	1Si(allyl)	1Ge(allyl)	1Sn(allyl)	allylSiPh ₃	allylSi(OC ₂ H ₄) ₃ N	allylSnPh ₃
HOMO /eV	-6.18	-6.02	-5.91	-6.60	-6.17	-6.37
NBO charge						
q(E)	+1.891	+1.697	+1.898	+1.872	+2.379	+1.852
q(C α)	-1.038	-0.970	-0.986	-1.009	-1.061	-0.958
q(C β)	-0.215	-0.222	-0.230	-0.227	-0.217	-0.244
q(C γ)	-0.492	-0.491	-0.506	-0.481	-0.491	-0.491
$\sigma(\text{E}-\text{C}\alpha) \rightarrow \pi^*(\text{C}\beta=\text{C}\gamma)$ /kcal/mol	3.84	5.29	9.37	5.61	5.90	9.53
$\Delta\delta_{\text{C}\beta-\text{C}\gamma}$ /ppm	25.3	24.4	28.4	18.7	27.8	23.5

B3PW91/DGDZVP (for Si, Ge, and Sn), 6-31+G**(for C, H, and N) level.

Figure S14. HOMO of (a) 1Si(allyl), (b) 1Ge(allyl), (c) 1Sn(allyl), (d) allylSiPh₃, (e) allylSi(OC₂H₄)₃N, and (f) allylSnPh₃ calculated at the B3PW91/DGDZVP (for Si, Ge, and Sn), 6-31+G** (for C, H, and N) level.

Figure S15. Natural bond orbitals of (a) 1Si(allyl), (b) 1Ge(allyl), (c) 1Sn(allyl), (d) allylSiPh₃, (e) allylSi(OC₂H₄)₃N, and (f) allylSnPh₃ calculated at the B3PW91/DGDZVP (for Si, Ge, and Sn), 6-31+G** (for C, H, and N) level.

More importantly, the origin of the anti-selectivity remains obscure. This is highlighted by the observation in Scheme 1b that both nucleophiles (allylSiMe₃ and Si(allyl)) yield the anti-selective product with cyclobutanone, suggesting the stereochemical outcome is not uniquely dictated by the nucleophile's properties as claimed. The steric effect can be a pivotal factor that affects the anti-selectivity. Consequently, without deeper mechanistic investigations, this work remains descriptive and fails to provide a coherent or convincing explanation for the observed reactivity and selectivity.

The issue that the reviewer pointed out is scientifically significant. We are very sorry for any potentially unclear discussions on this point, because the mechanism for the stereoselectivity in the allylation of cyclic ketones is not directly explained based on the Felkin-Anh model. In these reactions, the main interest is the steric demand of the allylsilatrane. To clarify this fact, we have modified the corresponding sentence in Scheme 1b, as shown below.

Furthermore, the steric effect of **1Si(allyl)** was evident in the allylation of α -oxy cyclobutanone **2q** (Scheme 1b). The resulting allylated product (**3q**) is an important synthetic intermediate for the substructure of a β -secretase modulator candidate.

Our responses to Referee 3 in the second round of revisions

We appreciate the reviewer's dedication.

First round of the review process

We appreciate the reviewer's dedication.

Our responses to Referee 4 in the second round of revisions

I am satisfied with the additions and corrections made by the authors in response to the bulk of my comments. However, I feel that the NMR spectra still have major deficiencies. When I initially commented “Some NMR spectra are scaled so small that they are almost unreadable (see S35, S45, S47, 48 for example). This also creates a problem in assessing impurities in the spectra. These NMR spectra definitely need to be expanded. Any prominent impurity should also be indicated and identified,” I did not mean to imply that these were the only problematic NMR spectra – the problem is rampant throughout the SI. I expected that the authors would critically reassess all spectra and expand the scale of others where obviously necessary. Compare the scale of the ¹H NMR spectra on S26, S27, S58, and S61 to the spectra on S29 and S31, for example. The scale for S29 and S31 is reasonable, but why are S26, S27, S58, and S61 scaled so small? Additionally, although the authors have expanded the vertical scale for the NMR spectra on S35, S45, S47, and S48, and have labeled the impurities with asterisks, the expanded spectra show levels of impurities that I would consider inappropriate for publication. If the authors could specifically identify the impurities as a solvent or diastereomer, the spectra could potentially be acceptable, but otherwise, these spectra contain too much unidentified impurity and would need to be re-purified (S45 and S47 are particularly problematic). As I mentioned in my initial evaluation, it is impossible to assess the purity of some of these compounds with their NMR spectra scaled so small. I suspect that several other compounds also contain impurity levels that exceed what is generally acceptable in a synthetic study.

We apologize for the shortcomings in the previous version of our work. We have now included expanded figures for S26, S27, S58, and S61 in the Supplementary Information (SI). The NMR spectra for S45 and S47 have been re-recorded following re-purification. Although the UV absorption of these products were insufficient to collect the desired fractions using column chromatography over silica gel, we have achieved some improvements in the purity of the isolated products. The updated charts have also been added to the SI.

Page S26: *anti-3d*

¹H NMR: (400 MHz, CDCl₃)

Page S27: *anti-3e*

¹H NMR: (400 MHz, CDCl₃)

Page S60 (Originally Page S58): *anti*-3aa

¹H NMR: (400 MHz, CDCl₃)

$^1\text{H NMR}$: (400 MHz, CDCl_3)

$^1\text{H NMR}$: (400 MHz, CDCl_3)

Asterisks represent inseparable impurities and residual solvents.

¹H NMR: (400 MHz, benzene-*d*₆)

First round of the review process

In the submitted manuscript, Yasuda and Konishi describe a strategy to selectively access the anti-diastereomer in the allylation of α -oxy ketones. This method addresses a difficult challenge in stereocontrolled additions to α -oxy ketones, whose chemistry is dominated by syn-diastereomer formation arising from chelation-controlled addition pathways. In this work, the authors expand upon their prior studies of Group 14 metallatrane derivatives, employing an allylcarbasilatrane derivative [1Si(allyl)] whose backbone includes aryl connectors between the metal center and the caged backbone. The authors demonstrate that a high degree of diastereocontrol can be achieved for the allylation of α -oxy ketones, using their carbasilatrane-derived nucleophile 1Si(allyl) to access the anti-diastereomer via a non-chelation-controlled mechanism and allylstannane/SnCl₂ to access the syn-diastereomer via a chelation-controlled mechanism. A key enabling factor involved in the selectivity achieved is the transannular nitrogen coordination to silicon, which results in the diminished Lewis acidity of Si while heightening its nucleophilicity relative to non-atrane-based allylsilanes, thus allowing access to a non-chelation-controlled allylation pathway. This process successfully generates the anti-diastereomer in the allylation of α -oxy aryl ketones with very high diastereoselectivity, though the use of α -oxy aldehydes results in poor selectivity, and the use of an α -oxy aliphatic ketone results in no product. Despite these substrate limitations, this is a nice demonstration of the unique reactivity that can be achieved through the variation of Group 14 metal and the atrane backbone. It not only provides a practical strategy to access anti-diastereomers from the allylation of α -oxy aldehydes, but also illustrates how the low Lewis acidity and tunable nucleophilicity of Group 14 metallatranes can allow access novel reaction pathways. I support this manuscript for publication in Nature Communication after my comments below are addressed.

We really appreciate the very positive comments.

The substrates in Table 2 primarily include α -aryl- α -oxy ketones. With the exception of the cyclic substrate 2j (entry 9), α -alkyl- α -oxy ketones are only used in examples exploring specialized substrates (bulky 2e, O–N-containing 2f) or in examples that fail (2k and 2l). I would like to see the inclusion of entries using an α -alkyl- α -acetoxo ketone and an α -alkyl- α -alkoxy ketone, which would make any potential limitations that might arise from the presence of an α -alkyl substituent more transparent.

We appreciate the kind suggestions. Although we have not attempted the allylations of α -alkyl- α -acetoxo ketones, the allylation of α -methyl- α -methoxy ketone **2j** using 1Si(allyl) and BF₃·OEt₂ has been demonstrated. The product **3j** was obtained in 97% yield with a high anti-selectivity (*syn/anti* = 4/96). The result has been added to entry 9 in Table 2. We have renumbered the compound numbers after adding α -methyl- α -methoxy ketone **2j** to Table 2.

The reaction also gave good yields and diastereoselectivities with *p*-chloro- or *p*-phenyl-substituted

benzoin methyl ethers (**2g**, **2h**) and a naphthyl ketone derivative (**2i**) (conditions A, Table 2, entries 6-8). Furthermore, the allylations of α -methyl- α -methoxy ketones **2j** and **2k** with a 2,5-disubstituted *trans*-tetrahydrofuran motif afforded **3j** and **3k** with nearly perfect *anti*-selectivity (*syn/anti* = 4/96 and 2/98) (conditions A, Table 2, entries 9–10). The selectivity of **3k** was dramatically improved relative to a previous study with allylMgCl (*syn/anti* = 29/71).⁴⁶

Entry 9 in Table 2.

The authors suggest that nucleophilicity of 1Si(allyl) is insufficient to react with the aliphatic ketone. This presents a fairly substantial limitation to the described method. Does the use of the more reactive 1Sn(allyl) result in any observed allylation? The authors should comment on the reactivity observed using 1Sn(allyl) with α -oxy aliphatic ketones.

We appreciate the kind suggestions. We have attempted the allylation of **2k** (now renamed **2l**) using **1Sn(allyl)** with $\text{BF}_3 \cdot \text{OEt}_2$, but no reaction occurred and the substrate was recovered quantitatively. Accordingly, we have added the following sentence to the main text.

Unfortunately, the desired product was not obtained from the allylation of α -oxy aliphatic ketone **2l** (conditions A, Table 2, entry 11), as the allylation of aliphatic ketones requires an allyl nucleophile with higher nucleophilicity; using **1Sn(allyl)** instead of **1Si(allyl)** had no effect.

According to the NBO data in Table S6, the C(γ) nucleophilicity of allylsilatrane is almost identical to that of 1Si(allyl). This suggests that the lack of stability of allylsilatrane in the presence of BF_3 is the sole reason why it is ineffective in this chemistry. This highlights the unique properties of 1Si(allyl) with respect to both stability and reactivity. This might be useful to mention in the text of the manuscript as it establishes a calculation-based comparison of reactivity between allylsilatrane and 1Si(allyl) that implies similar activating abilities. This is also noteworthy as allylcarbasilatrane would be expected to be significantly more nucleophilic than allylsilatrane, implying that the dampened electronic effect achieved by replacing the C(sp^3)-substituents in allylcarbasilatrane with the C(sp^2)-substituents of 1Si(allyl) closely parallels the effect achieved from the O substituents in allylsilatrane.

We appreciate the kind suggestions. According to the reviewer's suggestion, we have added the following explanations to the main text in order to highlight the unique properties of **1Si(allyl)**.

To investigate the electronic properties of **1E(allyl)**, we calculated its molecular orbitals (Figure 1c). The calculated energy of the HOMO of **1Si(allyl)** (−6.18 eV) suggests that its nucleophilicity is higher than that

of allylSiPh₃ **10** (-6.60 eV). The high nucleophilicity of **1Si**(allyl) is also supported by ¹³C NMR and natural-bond-orbital (NBO) analyses (Table S6). The difference in NBO charge between C_β and C_γ was higher for **1Si**(allyl) (|q(C_β)-q(C_γ)| = 0.277) than for **10** (|q(C_β)-q(C_γ)| = 0.254). Meanwhile, a second-order-perturbation analysis indicated that the hyperconjugation from the σ(Si-C_α) bond to the π*(C_β=C_γ) bond is less effective in **1Si**(allyl) (ΔE = 3.84 kcal/mol) than in **10** (ΔE = 5.61 kcal/mol), commensurate with weak electronic communication between the allylic moiety and the group-14-element center of **1Si**(allyl). Thus, the strong nucleophilic character of **1Si**(allyl) should reflect the charge localization rather than the stereoelectronic effects. **Notably, the results of the NBO analyses indicate that the nucleophilicity of 1Si(allyl) is similar to that of allylsilatrane 5, which rapidly decomposes under the applied reaction conditions. This highlights the balance of reactivity and stability in 1Si(allyl), facilitated by its rigid triaryl-based structure.** The stabilization of the silyl center by transannular interactions would also enhance the allyl-anion character. A crystallographic analysis confirmed that the molecular geometry of **1Si**(allyl) exhibits a nearly trigonal-bipyramidal structure with a five-coordinated silicon center.

Page 3 (bottom): I recommend that “which exhibits high nucleophilicity” be changed to “which exhibits heightened nucleophilicity” since this is in comparison to the allylsilane.

We appreciate the kind suggestion. We have changed the part in accordance with the reviewer’s suggestion.

Oxygen-ligated allylsilatrane allylSi(OC₂H₄)₃N **5**, **which exhibits heightened nucleophilicity**,^{37,38} was an ineffective allylation reagent (Table 1, entry 2). Nuclear-magnetic-resonance (NMR) measurements revealed that **5** decomposes rapidly in the presence of BF₃·Et₂O (Figure S10).

Page 4 (top): The line “The use of 6 in the presence of SnCl₂ and allylindium provided product syn-3a in high yield with excellent syn-selectivity...” makes it sound like 6 was treated with allylindium, which is not the case according to Table 1, entry 5. This line should be re-written, more clearly indicating that allylindium alone was employed as the nucleophile in the absence of 6 and SnCl₂.

We are very sorry for the unclear description. We have changed the part, as shown below.

Here, stannyl cation species generated by transmetalation with BF₃·Et₂O likely activate the substrate in a chelation-controlled manner.^{39,40} **Treating 2a with allylSnBu₃/SnCl₂⁴¹ or allylindium^{23,24,42} provided product syn-3a in high yield with excellent syn-selectivity (syn/anti = >99/1; Table 1, entries 4 and 5) via the chelation pathway.** The reactions with allylMgBr **8** or allylLi **9** proceeded without diastereoselectivity due to strong chelation combined with too high nucleophilicity (Table 1, entries 6 and 7).^{43,44}

Table S6: In the left column under NBO charge, q(Si) should be q(E) as Ge and Sn values are also

given.

We appreciate the reviewer's findings. We have corrected these parts.

Table S6. Summary of parameters related to nucleophilicity.

	1Si(allyl)	1Ge(allyl)	1Sn(allyl)	allylSiPh ₃	allylSi(OC ₂ H ₄) ₃ N	allylSnPh ₃
HOMO /eV	 -6.18	 -6.02	 -5.91	 -6.60	 -6.17	 -6.37
NBO charge						
q(E)	+1.891	+1.697	+1.898	+1.872	+2.379	+1.852
q(C _α)	-1.038	-0.970	-0.986	-1.009	-1.061	-0.958
q(C _β)	-0.215	-0.222	-0.230	-0.227	-0.217	-0.244
q(C _γ)	-0.492	-0.491	-0.506	-0.481	-0.491	-0.491
σ(E→C _α →π*(C _β -C _γ)/kcal/mol	3.84	5.29	9.37	5.61	5.90	9.53
Δδ _{Cβ-Cγ} /ppm	25.3	24.4	28.4	18.7	27.8	23.5

B3PW91/DGDZVP (for Si, Ge, and Sn), 6-31+G**(for C, H, and N) level.

Some NMR spectra are scaled so small that they are almost unreadable (see S35, S45, S47, 48 for example). This also creates a problem in assessing impurities in the spectra. These NMR spectra definitely need to be expanded. Any prominent impurity should also be indicated and identified.

We are very sorry for these shortcomings. We have added expanded figures with some notations for impurities to the SI.

pS35

^1H NMR: (400 MHz, CDCl_3)

Asterisks represent inseparable impurities and residual solvents.

pS45

^1H NMR: (400 MHz, CDCl_3)

Asterisks represent inseparable impurities and residual solvents.

pS47

¹H NMR: (400 MHz, benzene-d₆)

Asterisks represent inseparable impurities and residual solvents.

pS48

¹H NMR: (400 MHz, CDCl₃)

Asterisks represent inseparable impurities and residual solvents.

Our responses to Referee 2 in the 3rd round of revisions

In this third round of revision, the authors still have not addressed the fundamental mechanistic question of the anti-selectivity origin in the allylation of α -oxy ketones. Although the authors reiterate that stereoselectivity stems from the transition state conformations of α -oxy carbonyl compounds during nucleophilic attack, the manuscript continues to lack a substantive and direct investigation into this claim. Merely citing the polar Felkin-Ahn model and previous reports on α -oxy aldehydes is inadequate; it does not explain the unique stereocontrol mechanism in this new anti-selective allylation system. Since the core issue of stereoselectivity remains unresolved after two rounds of revision, the manuscript still lacks the compelling evidence and mechanistic depth required for publication in Nature Communications. I cannot recommend the publication of this work at the current stage.

We have performed DFT calculations to explain the observed *anti*-selective allylation. The results indicate that *anti*-selectivity arises through a non-chelation pathway. The calculated transition state leading to the *anti*-product is by 7.0 kcal/mol more stable than that leading to the *syn*-product. This increased stability was attributed to steric repulsion between the atrane moiety of **1Si(allyl)** and the phenyl group of **2a**.

During the evaluation of the reaction profiles, we determined that the transition state adopts a Cram-type conformation, wherein the α -methoxy group is located antiperiplanar to the carbonyl group of **2a**. This contrasts with the initially expected polar Felkin–Ahn model. The allylation of **2a** with the bulky **1Si(allyl)** and the small Lewis acid BF₃ may preferentially stabilize the Cram-type conformation. We thank the reviewer for this additional request, which has provided new insights into the transition-state conformations of the α -oxy carbonyl compound during the nucleophilic attack.

To present the theoretical results more clearly, we have revised the proposed mechanism in Figure 1d, updated the relevant sections of the main text, and included additional details in the Supporting Information (Chapter 9), which covers the computational estimation of the reaction profile. We thank the reviewer again for the valuable suggestion.

[Main text]

Figure 1. **a** Summary for the diastereoselective addition to α -chiral carbonyls. **b** *anti*-Selective allylation of α -oxy ketones by **1E(allyl)** ($E = \text{Si, Ge, or Sn}$; **this work**). **c** Calculated electronic properties of **1Si(allyl)** (left) and allylSiPh₃ **10** (right). **d** DFT-supported plausible reaction mechanism for the *anti*-selective allylation using **1Si(allyl)**.

DFT calculations at the B3LYP-D3/6-31G* level support the observed *anti*-selectivity via a non-chelation pathway (Figure S16). A plausible mechanism for the allylation of **2a** with allylsilatrane **1Si(allyl)** is shown in Figure 1d. First, ketone **2a** coordinates to BF₃. The activated ketone then interacts with **1Si(allyl)**, forming a carbon–carbon bond. The estimated transition state leading to the *anti*-product is by 7.0 kcal/mol more stable than that leading to the corresponding *syn*-product. This enhanced stability results from steric repulsion between the atrane moiety of **1Si(allyl)** and the phenyl group of **2a**. Interestingly, the conformation in the transition state resembles the Cram-type mode, in which the α -methoxy group is located antiperiplanar relative to the carbonyl group of **2a** activated by BF₃, rather than following the polar Felkin–Ahn model. The allylation of **2a** with the bulky **1Si(allyl)** and the small Lewis acid BF₃ is therefore likely to favor the Cram-type conformation over the initially anticipated polar Felkin–Ahn conformation. Subsequently, one of the fluorine atoms on the boron atom is captured by silyl cation **1Si**⁺, which results in the formation of the allyl product **Int** and **1SiF**. NMR monitoring confirmed the generation of **Int** and **1SiF** (Figure S12). Finally, protonation by quenching yields the homoallylic alcohol *anti*-**3a**. There is a possibility that an *in-situ* exchange of the allyl moiety between **1Si(allyl)** and BF₃·Et₂O occurs to form allylBF₂·Et₂O, albeit that a signal for allylBF₂·Et₂O was not observed in the NMR experiments. Therefore, it is considered that this *anti*-selective allylation proceeds via a non-chelation pathway. Further theoretical calculations to investigate the pathway in detail are currently in progress in our group. The high stability of **1Si(allyl)** derived from the atrane structure suppresses the undesired *in-situ* exchange process.

[SI]

9. Computational estimation of the reaction profile

9-1. General

All calculations were conducted using the Gaussian 16 Rev. C. 01 program.¹⁷ For the reaction profiles, the B3LYP-D3 functional was used in combination with the 6-31G* basis set. All molecular geometries were fully optimized at the singlet state and confirmed to have all positive vibrational frequencies for local minima or an imaginary vibrational frequency for transition states at the same theoretical level. By using the obtained optimized structures, Gibbs free energies including contribution of vibrational entropy at an appropriate temperature were described at the same theoretical level.

9-2. Summary for the reaction profiles for the allylations of **2a** with **1Si(allyl)** and **BF₃**

Table S13. Summary for the energy in the *anti*-allylation of **2a** calculated at the B3LYP-D3/6-31G* level.

state	total energy / Hartree	total energy + ZPE / Hartree	G (298 K) / Hartree	imaginary frequency / cm ⁻¹
2a ·BF ₃	-1272.7149136	0.4078174	-1272.3555980	–
1Si (allyl)	-1055.0570272	0.2669856	-1054.8411820	–
sum	-2327.7719408	0.6748030	-2327.1967800	–
TS (anti)	-2327.7816396	0.6789008	-2327.1760480	348.1523 i
Int' (anti)	-2327.7867023	0.6803922	-2327.1802890	–
Int (anti-3a)	-1072.5238490	0.3389021	-1072.2366570	–
1SiF	-1255.3173892	0.3396299	-1255.0229470	–
sum	-2327.8412382	0.6785320	-2327.2596040	–

Table S14. Summary for the energy in the *syn*-allylation of **2a** calculated at the B3LYP-D3/6-31G* level.

state	total energy / Hartree	total energy + ZPE / Hartree	G (298 K) / Hartree	imaginary frequency / cm ⁻¹
2a ·BF ₃	-1272.7149136	0.4078174	-1272.3555980	–
1Si (allyl)	-1055.0570272	0.2669856	-1054.8411820	–
sum	-2327.7719408	0.6748030	-2327.1967800	–
TS (syn)	-2327.7714492	0.6790139	-2327.1647800	350.2415 i
Int' (syn)	-2327.7788003	0.6806515	-2327.1705730	–
Int (syn-3a)	-1072.5164399	0.3390457	-1072.2288270	–
1SiF	-1255.3173892	0.3396299	-1255.0229470	–
sum	-2327.8338291	0.6786756	-2327.2517740	–

Figure S16. Energy diagram for the allylations of **2a** with **1Si(allyl)** and BF_3 . The Gibbs-free-energy values relative to **2a**· BF_3 and **1Si(allyl)** were calculated at the B3LYP-D3/6-31G* level.

9-3. Cartesian coordinates for optimized geometries

Table S15. Optimized Cartesian coordinates (Å) for **2a**· BF_3 in the singlet state at the B3LYP-D3/6-31G* level.

C	-0.35642500	-0.43376200	1.09910200	H	0.86090000	-2.86096700	1.55418400
C	-0.31207400	-1.66507000	0.19590300	H	-1.53336900	-0.74659500	-1.31970000
C	0.27519800	0.75228200	0.35670500	H	0.89931000	-4.84742700	0.07407500
C	0.35885400	-2.82672300	0.58928400	H	-1.49579200	-2.73388800	-2.80993400
C	-0.98715500	-1.63826600	-1.03167300	H	-3.33821800	-0.53003700	2.44037300
O	-1.65884200	-0.03804500	1.44708400	H	-1.85437300	-1.23483700	3.14819300
C	1.75334100	0.84734200	0.22131600	H	-2.54450600	-1.92253100	1.64836600
O	-0.42787400	1.61395500	-0.16785600	H	1.57827700	2.62816300	-0.96623800
C	0.37670800	-3.94840000	-0.24130800	H	2.26857700	-0.91839600	1.36929700
C	-0.97106800	-2.76055300	-1.85885100	H	-0.27366800	-4.78638000	-2.11815400
C	-2.37696300	-0.99275400	2.20839400	H	4.03640700	2.90362000	-1.25924700
C	2.27245700	1.92227500	-0.52307800	H	4.69326600	-0.64601300	1.08551500
C	2.64047200	-0.07530200	0.80021100	H	5.59379400	1.26064100	-0.22867100
C	-0.28563300	-3.91457700	-1.46959700	B	-2.35732500	2.17350400	-0.46453000
C	3.64566900	2.06993500	-0.68299400	F	-2.02017300	3.14353200	-1.31964900
C	4.01572900	0.07426400	0.63623100	F	-2.69313600	2.51453400	0.77799300
C	4.52050800	1.14586000	-0.10345900	F	-2.84907700	1.03726100	-0.98223600
H	0.23351200	-0.63740500	2.00868600				

Table S16. Optimized Cartesian coordinates (Å) for **1Si(allyl)** in the singlet state at the B3LYP-D3/6-31G* level.

Si	-0.12021100	0.07535600	0.59443400	C	-4.06070100	-1.64308800	-1.42055500
C	-0.61445400	0.45158700	2.43474000	H	-1.00394400	-0.46957200	2.88872200
C	1.25787700	-1.23745200	0.81218300	H	0.34094000	0.65403900	2.94561000
C	-1.56582700	1.57977300	2.71903500	H	-1.33312500	2.54006500	2.25890900
C	0.55342200	1.75487600	-0.03705500	H	0.79732000	-1.75143800	2.86259100
C	-1.71683600	-0.59652100	-0.21952100	H	-2.95332500	0.57157100	3.96881600
C	2.13649000	-1.53166800	-0.25382800	H	-3.30932700	2.35844100	3.64271000
C	1.44740800	-1.94128700	2.01405200	H	0.90753700	2.62963900	1.90981600
C	-2.66133400	1.50069500	3.48154100	H	-3.06699800	0.06908800	1.32976600
C	0.69857600	1.99288000	-1.42067100	H	2.55583400	0.16056000	-1.50890100
C	0.96513400	2.77695000	0.83580100	H	2.35091400	-1.34386800	-2.41243600
C	-1.66581000	-1.28556600	-1.45114600	H	3.80114700	-2.70218600	-0.95708700
C	-2.97755100	-0.45933300	0.38817700	H	2.56944700	-3.42897200	3.09834900
C	1.97348500	-0.76750600	-1.55204300	H	-0.75079600	0.92362900	-2.60167900
C	3.14444900	-2.48800000	-0.11578900	H	0.86000200	0.96336400	-3.33094900
C	2.45576600	-2.89721400	2.15703900	H	1.30895200	3.35705300	-2.97050300
C	0.32090500	0.87829600	-2.37311100	H	1.78565000	4.75693500	1.06749700
C	1.20066900	3.20480100	-1.89820500	H	0.12993600	-2.42240300	-1.74273300
C	1.47093200	3.99049600	0.36384300	H	-0.40291400	-1.57758100	-3.20036000
C	-0.31656200	-1.48838400	-2.10493000	H	-2.75318200	-2.31215800	-2.99974200
C	-2.82221900	-1.79520000	-2.04422400	H	-5.09592000	-0.84025900	0.29253100

Second round of the review process

I have reviewed the authors' response and the revised manuscript. While the use of second-order perturbation energy to explain nucleophilicity differences among allylic species of the same element (Table S6 in Response to Referees Letter) is reasonable, my central concern, as noted by another reviewer, regarding the unconvincing explanation for reaction selectivity remains unresolved.

Specifically, in Table 1, non-cage allylic Si/Sn compounds yield syn-type chelation products, whereas cage-type analogues afford anti-type non-chelation products. This product distribution cannot be rationalized solely by nucleophilicity, as the second-order perturbation energies (9.37 kcal/mol for 1Sn(allyl) vs. 5.61 kcal/mol for allylSiPh₃) do not align with the observed stereochemical outcomes. The reactivity shown in Scheme 1b further confirms that product selectivity is not governed exclusively by nucleophilicity.

Although the Group 14 metal-containing heteroatom cage derivatives developed here address the challenges associated with the anti-selective addition of α -oxy carbonyl compounds, elucidating the origin of anti-selective addition is essential to validate this new strategy. This reviewer strongly recommend that the authors provide conclusive computational evidence to explain the syn/anti-type chelation/non-chelation selectivity in the next revision.

We would like to clarify that the nucleophilicity estimated based on second-order perturbation energy is not suitable for explaining the stereoselectivity of allylations involving α -oxy ketones. In fact, we do not rely on nucleophilicity to account for the observed stereoselectivity. It is important to note here that the nucleophilicity of allylic reagents, which is derived from the electronic properties of the allyl moiety, does not have a direct connection to the stereoselectivity in their nucleophilic additions.

Instead, the stereoselectivity is determined by the transition-state conformations of α -oxy carbonyl compounds during the nucleophile attack. Based on seminal studies (for example, ref. 9), it has been empirically and theoretically accepted that α -oxy carbonyls induce *anti*-adducts in their allylations via a non-chelation pathway following the polar Felkin-Anh model. Reetz and Evans (see refs. 9, 31, and 33) have explained the *anti*-selectivity exhibited by α -oxy aldehydes due to their high electrophilicity. Given that a fundamental understanding of *syn/anti*-selectivity in nucleophilic additions to α -substituted carbonyls is well-established through abundant theoretical and experimental studies, we would like to argue that further computational evidence for our reactions is unnecessary.

In contrast to these pioneering studies, achieving *anti*-selective additions to α -oxy ketones—which are less reactive than the corresponding aldehydes as demonstrated by Reetz and Evans—remains elusive. Our manuscript addresses this critical issue in allylations. For the allylation of α -oxy ketones, it is essential to enhance either the electrophilicity of the ketone moiety or the nucleophilicity of the allylic reagent. However, both approaches encounter a significant and unavoidable challenge in achieving *anti*-selectivity. Traditionally, reagents with strong Lewis acidity are employed in these reactions; however, they can increase both the electrophilicity of the ketone group and the

nucleophilicity of the allylic reagent. Unfortunately, this often results in chelation-controlled additions that favor *syn*-selective outcomes.

To address this inherent issue and realize *anti*-selectivity in the allylation of α -oxy ketones, our goal is to harmonize high nucleophilicity with low Lewis acidity in an allylic reagent. This concept is central to our atrane-type reagent. To underscore this important point, we have added the following sentence to the main text.

that the molecular geometry of **1Si(allyl)** exhibits a nearly trigonal-bipyramidal structure with a five-coordinated silicon center. Most allylic reagents with high nucleophilicity feature a strong Lewis-acidic metal center, which facilitates the allylation of α -oxy ketones through a chelation pathway, resulting in *syn*-products.^{9,31,33} In contrast to the traditional methods for achieving *syn*-selective allylations, **1E(allyl)** enhances the nucleophilicity of the allylic group while simultaneously reducing the Lewis acidity of the metal center. This careful balance between high nucleophilicity and weak Lewis acidity in **1E(allyl)**, achieved through an atrane-type framework, allows for the distinctive *anti*-selective allylations of α -oxy ketones.

First round of the review process

The manuscript by Yasuda et al. developed an anti-selective alkenylation of α -oxy ketones using the previously reported Group-14 allyltranes (Chem Asian J. 2021, 16, 3118–3123). The authors employ computational investigations to compare the reactivity of atrane-type allyl nucleophiles with that of allylSiPh₃, suggesting that the former exhibits high nucleophilicity and low chelating ability. Despite the new approach to anti-selective allylation via a non-chelation pathway, the study is not acceptable for publication due to fundamental issues that remain unclear. Firstly, the reported second-order perturbation energy ($\sigma(\text{Sn}-\text{C}_\alpha) \rightarrow \pi^*(\text{C}_\beta=\text{C}_\gamma)$) is 9.37 kcal/mol for Sn(allyl) versus 5.61 kcal/mol for allylSiPh₃. However, a higher value indicates a stronger interaction, which is irreconcilable with the claim that transannular interactions stabilize the allyl-anion character. This contradiction invalidates the design of nucleophiles.

We are concerned that the reviewers may have misunderstood our findings. The extent of hyperconjugation, denoted as $\sigma(\text{E}-\text{C}_\alpha) \rightarrow \pi^*(\text{C}=\text{C})$, is more pronounced when the atomic number of element E increases. Consequently, when comparing homologs that contain identical substituents or ligands, the second-order perturbation energy tends to increase in the order of Si < Ge < Sn. However, the reviewer's point is that the comparison is between totally different allylic species with different frameworks. Thus, the pointed comparison is not suitable for discussion.

We have calculated the value for allylSnPh₃ to be 9.53 kcal/mol, which is higher than that for 1Sn(allyl) (9.37 kcal/mol). This estimation also supports our arguments. We have added the calculated values for allylSnPh₃ to Table S6, Figure S14, and Figure S15.

Table S6. Summary of parameters related to nucleophilicity.

	1Si(allyl)	1Ge(allyl)	1Sn(allyl)	allylSiPh ₃	allylSi(OC ₂ H ₄) ₃ N	allylSnPh ₃
HOMO /eV	-6.18	-6.02	-5.91	-6.60	-6.17	-6.37
NBO charge						
q(E)	+1.891	+1.697	+1.898	+1.872	+2.379	+1.852
q(C _α)	-1.038	-0.970	-0.986	-1.009	-1.061	-0.958
q(C _β)	-0.215	-0.222	-0.230	-0.227	-0.217	-0.244
q(C _γ)	-0.492	-0.491	-0.506	-0.481	-0.491	-0.491
$\sigma(\text{E}-\text{C}_\alpha) \rightarrow \pi^*(\text{C}_\beta=\text{C}_\gamma)$ /kcal/mol	3.84	5.29	9.37	5.61	5.90	9.53
$\Delta\delta_{\text{C}_\beta-\text{C}_\gamma}$ /ppm	25.3	24.4	28.4	18.7	27.8	23.5

B3PW91/DGDZVP (for Si, Ge, and Sn), 6-31+G**(for C, H, and N) level.

Figure S14. HOMO of (a) 1Si(allyl), (b) 1Ge(allyl), (c) 1Sn(allyl), (d) allylSiPh₃, (e) allylSi(OC₂H₄)₃N, and (f) allylSnPh₃ calculated at the B3PW91/DGDZVP (for Si, Ge, and Sn), 6-31+G** (for C, H, and N) level.

Figure S15. Natural bond orbitals of (a) 1Si(allyl), (b) 1Ge(allyl), (c) 1Sn(allyl), (d) allylSiPh₃, (e) allylSi(OC₂H₄)₃N, and (f) allylSnPh₃ calculated at the B3PW91/DGDZVP (for Si, Ge, and Sn), 6-31+G** (for C, H, and N) level.

More importantly, the origin of the anti-selectivity remains obscure. This is highlighted by the observation in Scheme 1b that both nucleophiles (allylSiMe₃ and Si(allyl)) yield the anti-selective product with cyclobutanone, suggesting the stereochemical outcome is not uniquely dictated by the nucleophile's properties as claimed. The steric effect can be a pivotal factor that affects the anti-selectivity. Consequently, without deeper mechanistic investigations, this work remains descriptive and fails to provide a coherent or convincing explanation for the observed reactivity and selectivity.

The issue that the reviewer pointed out is scientifically significant. We are very sorry for any potentially unclear discussions on this point, because the mechanism for the stereoselectivity in the allylation of cyclic ketones is not directly explained based on the Felkin-Anh model. In these reactions, the main interest is the steric demand of the allylsilatrane. To clarify this fact, we have modified the corresponding sentence in Scheme 1b, as shown below.

Furthermore, the steric effect of **1**Si(allyl) was evident in the allylation of α -oxy cyclobutanone **2q** (Scheme 1b). The resulting allylated product (**3q**) is an important synthetic intermediate for the substructure of a β -secretase modulator candidate.

Our responses to Referee 4 in the 3rd round of revisions

The NMR spectra contained in the SI are significantly improved. I now find this manuscript suitable for publication in Nat. Comm.

We appreciate the reviewer's acceptance of our revisions. The reviewer's suggestions have undoubtedly improved our manuscript.

Second round of the review process

I am satisfied with the additions and corrections made by the authors in response to the bulk of my comments. However, I feel that the NMR spectra still have major deficiencies. When I initially commented “Some NMR spectra are scaled so small that they are almost unreadable (see S35, S45, S47, 48 for example). This also creates a problem in assessing impurities in the spectra. These NMR spectra definitely need to be expanded. Any prominent impurity should also be indicated and identified,” I did not mean to imply that these were the only problematic NMR spectra – the problem is rampant throughout the SI. I expected that the authors would critically reassess all spectra and expand the scale of others where obviously necessary. Compare the scale of the ¹H NMR spectra on S26, S27, S58, and S61 to the spectra on S29 and S31, for example. The scale for S29 and S31 is reasonable, but why are S26, S27, S58, and S61 scaled so small? Additionally, although the authors have expanded the vertical scale for the NMR spectra on S35, S45, S47, and S48, and have labeled the impurities with asterisks, the expanded spectra show levels of impurities that I would consider inappropriate for publication. If the authors could specifically identify the impurities as a solvent or diastereomer, the spectra could potentially be acceptable, but otherwise, these spectra contain too much unidentified impurity and would need to be re-purified (S45 and S47 are particularly problematic). As I mentioned in my initial evaluation, it is impossible to assess the purity of some of these compounds with their NMR spectra scaled so small. I suspect that several other compounds also contain impurity levels that exceed what is generally acceptable in a synthetic study.

We apologize for the shortcomings in the previous version of our work. We have now included expanded figures for S26, S27, S58, and S61 in the Supplementary Information (SI). The NMR spectra for S45 and S47 have been re-recorded following re-purification. Although the UV absorption of these products were insufficient to collect the desired fractions using column chromatography over silica gel, we have achieved some improvements in the purity of the isolated products. The updated charts have also been added to the SI.

Page S26: *anti-3d*

$^1\text{H NMR}$: (400 MHz, CDCl_3)

Page S27: *anti-3e*

$^1\text{H NMR}$: (400 MHz, CDCl_3)

Page S60 (Originally Page S58): *anti*-3aa

¹H NMR: (400 MHz, CDCl₃)

¹H NMR: (400 MHz, CDCl₃)

¹H NMR: (400 MHz, CDCl₃)

Asterisks represent inseparable impurities and residual solvents.

¹H NMR: (400 MHz, benzene-*d*₆)

First round of the review process

In the submitted manuscript, Yasuda and Konishi describe a strategy to selectively access the anti-diastereomer in the allylation of α -oxy ketones. This method addresses a difficult challenge in stereocontrolled additions to α -oxy ketones, whose chemistry is dominated by syn-diastereomer formation arising from chelation-controlled addition pathways. In this work, the authors expand upon their prior studies of Group 14 metallatrane derivatives, employing an allylcarbasilatrane derivative [1Si(allyl)] whose backbone includes aryl connectors between the metal center and the caged backbone. The authors demonstrate that a high degree of diastereocontrol can be achieved for the allylation of α -oxy ketones, using their carbasilatrane-derived nucleophile 1Si(allyl) to access the anti-diastereomer via a non-chelation-controlled mechanism and allylstannane/SnCl₂ to access the syn-diastereomer via a chelation-controlled mechanism. A key enabling factor involved in the selectivity achieved is the transannular nitrogen coordination to silicon, which results in the diminished Lewis acidity of Si while heightening its nucleophilicity relative to non-atrane-based allylsilanes, thus allowing access to a non-chelation-controlled allylation pathway. This process successfully generates the anti-diastereomer in the allylation of α -oxy aryl ketones with very high diastereoselectivity, though the use of α -oxy aldehydes results in poor selectivity, and the use of an α -oxy aliphatic ketone results in no product. Despite these substrate limitations, this is a nice demonstration of the unique reactivity that can be achieved through the variation of Group 14 metal and the atrane backbone. It not only provides a practical strategy to access anti-diastereomers from the allylation of α -oxy aldehydes, but also illustrates how the low Lewis acidity and tunable nucleophilicity of Group 14 metallatranes can allow access novel reaction pathways. I support this manuscript for publication in Nature Communication after my comments below are addressed.

We really appreciate the very positive comments.

The substrates in Table 2 primarily include α -aryl- α -oxy ketones. With the exception of the cyclic substrate 2j (entry 9), α -alkyl- α -oxy ketones are only used in examples exploring specialized substrates (bulky 2e, O–N-containing 2f) or in examples that fail (2k and 2l). I would like to see the inclusion of entries using an α -alkyl- α -acetoxo ketone and an α -alkyl- α -alkoxy ketone, which would make any potential limitations that might arise from the presence of an α -alkyl substituent more transparent.

We appreciate the kind suggestions. Although we have not attempted the allylations of α -alkyl- α -acetoxo ketones, the allylation of α -methyl- α -methoxy ketone **2j** using 1Si(allyl) and BF₃·OEt₂ has been demonstrated. The product **3j** was obtained in 97% yield with a high anti-selectivity (*syn/anti* = 4/96). The result has been added to entry 9 in Table 2. We have renumbered the compound numbers after adding α -methyl- α -methoxy ketone **2j** to Table 2.

The reaction also gave good yields and diastereoselectivities with *p*-chloro- or *p*-phenyl-substituted

benzoin methyl ethers (**2g**, **2h**) and a naphthyl ketone derivative (**2i**) (conditions A, Table 2, entries 6-8). Furthermore, the allylations of α -methyl- α -methoxy ketones **2j** and **2k** with a 2,5-disubstituted *trans*-tetrahydrofuran motif afforded **3j** and **3k** with nearly perfect *anti*-selectivity (*syn/anti* = 4/96 and 2/98) (conditions A, Table 2, entries 9–10). The selectivity of **3k** was dramatically improved relative to a previous study with allylMgCl (*syn/anti* = 29/71).⁴⁶

Entry 9 in Table 2.

The authors suggest that nucleophilicity of 1Si(allyl) is insufficient to react with the aliphatic ketone. This presents a fairly substantial limitation to the described method. Does the use of the more reactive 1Sn(allyl) result in any observed allylation? The authors should comment on the reactivity observed using 1Sn(allyl) with α -oxy aliphatic ketones.

We appreciate the kind suggestions. We have attempted the allylation of **2k** (now renamed **2l**) using **1Sn(allyl)** with $\text{BF}_3 \cdot \text{OEt}_2$, but no reaction occurred and the substrate was recovered quantitatively. Accordingly, we have added the following sentence to the main text.

Unfortunately, the desired product was not obtained from the allylation of α -oxy aliphatic ketone **2l** (conditions A, Table 2, entry 11), as the allylation of aliphatic ketones requires an allyl nucleophile with higher nucleophilicity; using **1Sn(allyl)** instead of **1Si(allyl)** had no effect.

According to the NBO data in Table S6, the C(γ) nucleophilicity of allylsilatrane is almost identical to that of 1Si(allyl). This suggests that the lack of stability of allylsilatrane in the presence of BF_3 is the sole reason why it is ineffective in this chemistry. This highlights the unique properties of 1Si(allyl) with respect to both stability and reactivity. This might be useful to mention in the text of the manuscript as it establishes a calculation-based comparison of reactivity between allylsilatrane and 1Si(allyl) that implies similar activating abilities. This is also noteworthy as allylcarbasilatrane would be expected to be significantly more nucleophilic than allylsilatrane, implying that the dampened electronic effect achieved by replacing the C(sp^3)-substituents in allylcarbasilatrane with the C(sp^2)-substituents of 1Si(allyl) closely parallels the effect achieved from the O substituents in allylsilatrane.

We appreciate the kind suggestions. According to the reviewer's suggestion, we have added the following explanations to the main text in order to highlight the unique properties of **1Si(allyl)**.

To investigate the electronic properties of **1E(allyl)**, we calculated its molecular orbitals (Figure 1c). The calculated energy of the HOMO of **1Si(allyl)** (−6.18 eV) suggests that its nucleophilicity is higher than that

of allylSiPh₃ **10** (-6.60 eV). The high nucleophilicity of **1Si**(allyl) is also supported by ¹³C NMR and natural-bond-orbital (NBO) analyses (Table S6). The difference in NBO charge between C_β and C_γ was higher for **1Si**(allyl) ($|q(C_{\beta})-q(C_{\gamma})| = 0.277$) than for **10** ($|q(C_{\beta})-q(C_{\gamma})| = 0.254$). Meanwhile, a second-order-perturbation analysis indicated that the hyperconjugation from the $\sigma(\text{Si}-\text{C}_{\alpha})$ bond to the $\pi^*(\text{C}_{\beta}=\text{C}_{\gamma})$ bond is less effective in **1Si**(allyl) ($\Delta E = 3.84$ kcal/mol) than in **10** ($\Delta E = 5.61$ kcal/mol), commensurate with weak electronic communication between the allylic moiety and the group-14-element center of **1Si**(allyl). Thus, the strong nucleophilic character of **1Si**(allyl) should reflect the charge localization rather than the stereoelectronic effects. **Notably, the results of the NBO analyses indicate that the nucleophilicity of **1Si**(allyl) is similar to that of allylsilatrane **5**, which rapidly decomposes under the applied reaction conditions. This highlights the balance of reactivity and stability in **1Si**(allyl), facilitated by its rigid triaryl-based structure.** The stabilization of the silyl center by transannular interactions would also enhance the allyl-anion character. A crystallographic analysis confirmed that the molecular geometry of **1Si**(allyl) exhibits a nearly trigonal-bipyramidal structure with a five-coordinated silicon center.

Page 3 (bottom): I recommend that “which exhibits high nucleophilicity” be changed to “which exhibits heightened nucleophilicity” since this is in comparison to the allylsilane.

We appreciate the kind suggestion. We have changed the part in accordance with the reviewer’s suggestion.

Oxygen-ligated allylsilatrane allylSi(OC₂H₄)₃N **5**, **which exhibits heightened nucleophilicity**,^{37,38} was an ineffective allylation reagent (Table 1, entry 2). Nuclear-magnetic-resonance (NMR) measurements revealed that **5** decomposes rapidly in the presence of BF₃·Et₂O (Figure S10).

*Page 4 (top): The line “The use of **6** in the presence of SnCl₂ and allylindium provided product *syn*-**3a** in high yield with excellent *syn*-selectivity...” makes it sound like **6** was treated with allylindium, which is not the case according to Table 1, entry 5. This line should be re-written, more clearly indicating that allylindium alone was employed as the nucleophile in the absence of **6** and SnCl₂.*

We are very sorry for the unclear description. We have changed the part, as shown below.

Here, stannyl cation species generated by transmetalation with BF₃·Et₂O likely activate the substrate in a chelation-controlled manner.^{39,40} **Treating **2a** with allylSnBu₃/SnCl₂⁴¹ or allylindium^{23,24,42} provided product *syn*-**3a** in high yield with excellent *syn*-selectivity (*syn/anti* = >99/1; Table 1, entries 4 and 5) via the chelation pathway.** The reactions with allylMgBr **8** or allylLi **9** proceeded without diastereoselectivity due to strong chelation combined with too high nucleophilicity (Table 1, entries 6 and 7).^{43,44}

Table S6: In the left column under NBO charge, $q(\text{Si})$ should be $q(\text{E})$ as Ge and Sn values are also

given.

We appreciate the reviewer's findings. We have corrected these parts.

Table S6. Summary of parameters related to nucleophilicity.

	1Si(allyl)	1Ge(allyl)	1Sn(allyl)	allylSiPh ₃	allylSi(OC ₂ H ₄) ₃ N	allylSnPh ₃
HOMO /eV	 -6.18	 -6.02	 -5.91	 -6.60	 -6.17	 -6.37
NBO charge						
q(E)	+1.891	+1.697	+1.898	+1.872	+2.379	+1.852
q(C _α)	-1.038	-0.970	-0.986	-1.009	-1.061	-0.958
q(C _β)	-0.215	-0.222	-0.230	-0.227	-0.217	-0.244
q(C _γ)	-0.492	-0.491	-0.506	-0.481	-0.491	-0.491
σ(E→C _α →π*(C _β -C _γ)/kcal/mol	3.84	5.29	9.37	5.61	5.90	9.53
Δδ _{Cβ-Cγ} /ppm	25.3	24.4	28.4	18.7	27.8	23.5

B3PW91/DGDZVP (for Si, Ge, and Sn), 6-31+G**(for C, H, and N) level.

Some NMR spectra are scaled so small that they are almost unreadable (see S35, S45, S47, 48 for example). This also creates a problem in assessing impurities in the spectra. These NMR spectra definitely need to be expanded. Any prominent impurity should also be indicated and identified.

We are very sorry for these shortcomings. We have added expanded figures with some notations for impurities to the SI.

pS35

¹H NMR: (400 MHz, CDCl₃)

Asterisks represent inseparable impurities and residual solvents.

pS45

¹H NMR: (400 MHz, CDCl₃)

Asterisks represent inseparable impurities and residual solvents.

pS47

^1H NMR: (400 MHz, benzene- d_6)

Asterisks represent inseparable impurities and residual solvents.

pS48

^1H NMR: (400 MHz, CDCl_3)

Asterisks represent inseparable impurities and residual solvents.

Our responses to Referee 2 in the 4th round of revisions

In this revised version, the authors have provided DFT calculations to clarify the origin of anti-selectivity in the allylation reaction and the syn/anti selectivity of the nucleophilic attack by α -oxy carbonyl compounds. However, the Gibbs free energy data were obtained under gas-phase conditions. In real reaction systems, solvent effects can significantly influence the energy trends and alter the interpretation of selectivity. The energy obtained in the solvent based on the structures optimized in gas-phase should be used for discussion in the main text. This work can be accepted for publication after the authors update the energy data.

We appreciate the reviewer's suggestion. We have updated the energy data to include solvent effects calculated at the B3LYP-D3/6-31G*/SMD (dichloromethane) level. The energy profile in the solvent supports the observed *anti*-selectivity. The revised energy data have been incorporated into both the main text and the SI.

[Main text]

DFT calculations at the B3LYP-D3/6-31G*/SMD (dichloromethane)//B3LYP-D3/6-31G* level support the observed *anti*-selectivity via a non-chelation pathway (Figure S16). A plausible mechanism for the allylation of **2a** with allylsilatrane **1Si(allyl)** is shown in Figure 1d. First, ketone **2a** coordinates to BF₃. The activated ketone then interacts with **1Si(allyl)**, forming a carbon–carbon bond. The estimated transition state leading to the *anti*-product is by 6.4 kcal/mol more stable than that leading to the corresponding *syn*-product. This enhanced stability results from steric repulsion between the atrane moiety of **1Si(allyl)** and the phenyl group of **2a**. Interestingly, the conformation in the transition state resembles the Cram-type mode, in which the α -methoxy group is located antiperiplanar relative to the carbonyl group of **2a** activated by BF₃, rather than following the polar Felkin–Ahn model. The allylation of **2a** with the bulky **1Si(allyl)** and the small Lewis acid BF₃ is therefore likely to favor the Cram-type conformation over the initially anticipated polar Felkin–Ahn conformation. Subsequently, one of the fluorine atoms on the boron atom is captured by silyl cation **1Si⁺**, which results in the formation of the allyl product **Int** and **1SiF**. NMR monitoring confirmed the generation of **Int** and **1SiF** (Figure S12). Finally, protonation by quenching yields the homoallylic alcohol *anti*-**3a**. There is a possibility that an *in-situ* exchange of the allyl moiety between **1Si(allyl)** and BF₃·Et₂O occurs to form allylBF₂·Et₂O, albeit that a signal for allylBF₂·Et₂O was not observed in the NMR experiments. Therefore, it is considered that this *anti*-selective allylation proceeds via a non-chelation pathway. Further theoretical calculations to investigate the pathway in detail are currently in progress in our group. The high stability of **1Si(allyl)** derived from the atrane structure suppresses the undesired *in-situ* exchange process.

[SI]

9-1. General

All calculations were conducted using the Gaussian 16 Rev. C. 01 program.¹⁷ For the reaction profiles, the B3LYP-D3 functional was used in combination with the 6-31G* basis set. All molecular geometries were fully optimized at the singlet state and confirmed to have all positive vibrational frequencies for local minima or an imaginary vibrational frequency for transition states at the same theoretical level. By using the obtained optimized structures, Gibbs free energies including contribution of vibrational entropy at an appropriate temperature were described at the same theoretical level, in which solvation effect was introduced using the SMD model (dichloromethane).

9-2. Summary for the reaction profiles for the allylations of 2a with 1Si(allyl) and BF₃

Table S13. Summary for the energy in the *anti*-allylation of 2a

state	total energy / Hartree ^{a)}	Zero-point energy / Hartree ^{b)}	Thermal correlation for Gibbs energy / Hartree ^{b)}	Total energy +ZPE / Hartree	Gibus energy (298 K) / Hartree	imaginary frequency / cm ⁻¹ ^{b)}
2a·BF ₃	-1055.0808175	0.2669856	0.215846	-1054.8138319	-1054.8649715	–
1Si(allyl)	-1272.7421729	0.4078174	0.3593160	-1272.3343555	-1272.3828569	–
sum	-2327.8229904	0.6748030	0.5751620	-2327.1481874	-2327.2478284	–
TS(anti)	-2327.8368437	0.6789008	0.6055910	-2327.1579429	-2327.2312527	348.1523 i
Int'(anti)	-2327.8543954	0.6803922	0.6064130	-2327.1740032	-2327.2479824	–
Int(anti -3a)	-1072.5440874	0.3389021	0.2871920	-1072.2051853	-1072.2568954	–
1SiF	-1255.3459658	0.3396299	0.2944420	-1255.0063359	-1255.0515238	–
sum	-2327.8900532	0.6785320	0.5816340	-2327.2115212	-2327.3084192	–

^{a)} B3LYP-D3/6-31G*/SMD (dichloromethane)//B3LYP-D3/6-31G* ^{b)} B3LYP-D3/6-31G*

Table S14. Summary for the energy in the *syn*-allylation of 2a.

state	total energy / Hartree ^{a)}	Zero-point energy / Hartree ^{b)}	Thermal correlation for Gibbs energy / Hartree ^{b)}	Total energy +ZPE / Hartree	Gibus energy (298 K) / Hartree	imaginary frequency / cm ⁻¹ ^{b)}
2a·BF ₃	-1055.0808175	0.2669856	0.215846	-1054.8138319	-1054.8649715	–
1Si(allyl)	-1272.7421729	0.4078174	0.3593160	-1272.3343555	-1272.3828569	–
sum	-2327.8229904	0.6748030	0.5751620	-2327.1481874	-2327.2478284	–
TS(syn)	-2327.8276692	0.6790139	0.6066700	-2327.1486553	-2327.2209992	350.2415 i
Int'(syn)	-2327.8509799	0.6806515	0.6082270	-2327.1703284	-2327.2427529	–
Int(syn -3a)	-1072.5369396	0.3396299	0.2944420	-1072.1973097	-1072.2424976	–
1SiF	-1255.3459658	0.3396299	0.2944420	-1255.0063359	-1255.0515238	–
sum	-2327.8829054	0.6792598	0.5888840	-2327.2036456	-2327.2940214	–

^{a)} B3LYP-D3/6-31G*/SMD (dichloromethane)//B3LYP-D3/6-31G* ^{b)} B3LYP-D3/6-31G*

Figure S16. Energy diagram for the allylations of **2a** with **1Si(allyl)** and BF_3 . The Gibbs-free-energy values relative to **2a**· BF_3 and **1Si(allyl)** were calculated at the B3LYP-D3/6-31G*/SMD (dichloromethane)//B3LYP-D3/6-31G* level.

Our responses to Referee 2 in the 3rd round of revisions

In this third round of revision, the authors still have not addressed the fundamental mechanistic question of the anti-selectivity origin in the allylation of α -oxy ketones. Although the authors reiterate that stereoselectivity stems from the transition state conformations of α -oxy carbonyl compounds during nucleophilic attack, the manuscript continues to lack a substantive and direct investigation into this claim. Merely citing the polar Felkin-Ahn model and previous reports on α -oxy aldehydes is inadequate; it does not explain the unique stereocontrol mechanism in this new anti-selective allylation system. Since the core issue of stereoselectivity remains unresolved after two rounds of revision, the manuscript still lacks the compelling evidence and mechanistic depth required for publication in Nature Communications. I cannot recommend the publication of this work at the current stage.

We have performed DFT calculations to explain the observed *anti*-selective allylation. The results indicate that *anti*-selectivity arises through a non-chelation pathway. The calculated transition state leading to the *anti*-product is by 7.0 kcal/mol more stable than that leading to the *syn*-product. This increased stability was attributed to steric repulsion between the atrane moiety of **1Si(allyl)** and the phenyl group of **2a**.

During the evaluation of the reaction profiles, we determined that the transition state adopts a Cram-type conformation, wherein the α -methoxy group is located antiperiplanar to the carbonyl group of **2a**. This contrasts with the initially expected polar Felkin–Ahn model. The allylation of **2a** with the bulky **1Si(allyl)** and the small Lewis acid BF_3 may preferentially stabilize the Cram-type conformation. We thank the reviewer for this additional request, which has provided new insights into the transition-state conformations of the α -oxy carbonyl compound during the nucleophilic attack.

To present the theoretical results more clearly, we have revised the proposed mechanism in Figure 1d, updated the relevant sections of the main text, and included additional details in the Supporting Information (Chapter 9), which covers the computational estimation of the reaction profile. We thank the reviewer again for the valuable suggestion.

[Main text]

Figure 1. **a** Summary for the diastereoselective addition to α -chiral carbonyls. **b** *anti*-Selective allylation of α -oxy ketones by **1E(allyl)** ($E = \text{Si, Ge, or Sn}$; **this work**). **c** Calculated electronic properties of **1Si(allyl)** (left) and allylSiPh₃ **10** (right). **d** DFT-supported plausible reaction mechanism for the *anti*-selective allylation using **1Si(allyl)**.

DFT calculations at the B3LYP-D3/6-31G* level support the observed *anti*-selectivity via a non-chelation pathway (Figure S16). A plausible mechanism for the allylation of **2a** with allylsilatrane **1Si(allyl)** is shown in Figure 1d. First, ketone **2a** coordinates to BF₃. The activated ketone then interacts with **1Si(allyl)**, forming a carbon–carbon bond. The estimated transition state leading to the *anti*-product is by 7.0 kcal/mol more stable than that leading to the corresponding *syn*-product. This enhanced stability results from steric repulsion between the atrane moiety of **1Si(allyl)** and the phenyl group of **2a**. Interestingly, the conformation in the transition state resembles the Cram-type mode, in which the α -methoxy group is located antiperiplanar relative to the carbonyl group of **2a** activated by BF₃, rather than following the polar Felkin–Ahn model. The allylation of **2a** with the bulky **1Si(allyl)** and the small Lewis acid BF₃ is therefore likely to favor the Cram-type conformation over the initially anticipated polar Felkin–Ahn conformation. Subsequently, one of the fluorine atoms on the boron atom is captured by silyl cation **1Si**⁺, which results in the formation of the allyl product **Int** and **1SiF**. NMR monitoring confirmed the generation of **Int** and **1SiF** (Figure S12). Finally, protonation by quenching yields the homoallylic alcohol *anti*-**3a**. There is a possibility that an *in-situ* exchange of the allyl moiety between **1Si(allyl)** and BF₃·Et₂O occurs to form allylBF₂·Et₂O, albeit that a signal for allylBF₂·Et₂O was not observed in the NMR experiments. Therefore, it is considered that this *anti*-selective allylation proceeds via a non-chelation pathway. Further theoretical calculations to investigate the pathway in detail are currently in progress in our group. The high stability of **1Si(allyl)** derived from the atrane structure suppresses the undesired *in-situ* exchange process.

[SI]

9. Computational estimation of the reaction profile

9-1. General

All calculations were conducted using the Gaussian 16 Rev. C. 01 program.¹⁷ For the reaction profiles, the B3LYP-D3 functional was used in combination with the 6-31G* basis set. All molecular geometries were fully optimized at the singlet state and confirmed to have all positive vibrational frequencies for local minima or an imaginary vibrational frequency for transition states at the same theoretical level. By using the obtained optimized structures, Gibbs free energies including contribution of vibrational entropy at an appropriate temperature were described at the same theoretical level.

9-2. Summary for the reaction profiles for the allylations of **2a** with **1Si(allyl)** and **BF₃**

Table S13. Summary for the energy in the *anti*-allylation of **2a** calculated at the B3LYP-D3/6-31G* level.

state	total energy / Hartree	total energy + ZPE / Hartree	G (298 K) / Hartree	imaginary frequency / cm ⁻¹
2a ·BF ₃	-1272.7149136	0.4078174	-1272.3555980	–
1Si(allyl)	-1055.0570272	0.2669856	-1054.8411820	–
sum	-2327.7719408	0.6748030	-2327.1967800	–
TS(anti)	-2327.7816396	0.6789008	-2327.1760480	348.1523 i
Int'(anti)	-2327.7867023	0.6803922	-2327.1802890	–
Int(anti-3a)	-1072.5238490	0.3389021	-1072.2366570	–
1SiF	-1255.3173892	0.3396299	-1255.0229470	–
sum	-2327.8412382	0.6785320	-2327.2596040	–

Table S14. Summary for the energy in the *syn*-allylation of **2a** calculated at the B3LYP-D3/6-31G* level.

state	total energy / Hartree	total energy + ZPE / Hartree	G (298 K) / Hartree	imaginary frequency / cm ⁻¹
2a ·BF ₃	-1272.7149136	0.4078174	-1272.3555980	–
1Si(allyl)	-1055.0570272	0.2669856	-1054.8411820	–
sum	-2327.7719408	0.6748030	-2327.1967800	–
TS(syn)	-2327.7714492	0.6790139	-2327.1647800	350.2415 i
Int'(syn)	-2327.7788003	0.6806515	-2327.1705730	–
Int(syn-3a)	-1072.5164399	0.3390457	-1072.2288270	–
1SiF	-1255.3173892	0.3396299	-1255.0229470	–
sum	-2327.8338291	0.6786756	-2327.2517740	–

Figure S16. Energy diagram for the allylations of **2a** with **1Si(allyl)** and BF_3 . The Gibbs-free-energy values relative to **2a**· BF_3 and **1Si(allyl)** were calculated at the B3LYP-D3/6-31G* level.

9-3. Cartesian coordinates for optimized geometries

Table S15. Optimized Cartesian coordinates (Å) for **2a**· BF_3 in the singlet state at the B3LYP-D3/6-31G* level.

C	-0.35642500	-0.43376200	1.09910200	H	0.86090000	-2.86096700	1.55418400
C	-0.31207400	-1.66507000	0.19590300	H	-1.53336900	-0.74659500	-1.31970000
C	0.27519800	0.75228200	0.35670500	H	0.89931000	-4.84742700	0.07407500
C	0.35885400	-2.82672300	0.58928400	H	-1.49579200	-2.73388800	-2.80993400
C	-0.98715500	-1.63826600	-1.03167300	H	-3.33821800	-0.53003700	2.44037300
O	-1.65884200	-0.03804500	1.44708400	H	-1.85437300	-1.23483700	3.14819300
C	1.75334100	0.84734200	0.22131600	H	-2.54450600	-1.92253100	1.64836600
O	-0.42787400	1.61395500	-0.16785600	H	1.57827700	2.62816300	-0.96623800
C	0.37670800	-3.94840000	-0.24130800	H	2.26857700	-0.91839600	1.36929700
C	-0.97106800	-2.76055300	-1.85885100	H	-0.27366800	-4.78638000	-2.11815400
C	-2.37696300	-0.99275400	2.20839400	H	4.03640700	2.90362000	-1.25924700
C	2.27245700	1.92227500	-0.52307800	H	4.69326600	-0.64601300	1.08551500
C	2.64047200	-0.07530200	0.80021100	H	5.59379400	1.26064100	-0.22867100
C	-0.28563300	-3.91457700	-1.46959700	B	-2.35732500	2.17350400	-0.46453000
C	3.64566900	2.06993500	-0.68299400	F	-2.02017300	3.14353200	-1.31964900
C	4.01572900	0.07426400	0.63623100	F	-2.69313600	2.51453400	0.77799300
C	4.52050800	1.14586000	-0.10345900	F	-2.84907700	1.03726100	-0.98223600
H	0.23351200	-0.63740500	2.00868600				

Table S16. Optimized Cartesian coordinates (Å) for **1Si(allyl)** in the singlet state at the B3LYP-D3/6-31G* level.

Si	-0.12021100	0.07535600	0.59443400	C	-4.06070100	-1.64308800	-1.42055500
C	-0.61445400	0.45158700	2.43474000	H	-1.00394400	-0.46957200	2.88872200
C	1.25787700	-1.23745200	0.81218300	H	0.34094000	0.65403900	2.94561000
C	-1.56582700	1.57977300	2.71903500	H	-1.33312500	2.54006500	2.25890900
C	0.55342200	1.75487600	-0.03705500	H	0.79732000	-1.75143800	2.86259100
C	-1.71683600	-0.59652100	-0.21952100	H	-2.95332500	0.57157100	3.96881600
C	2.13649000	-1.53166800	-0.25382800	H	-3.30932700	2.35844100	3.64271000
C	1.44740800	-1.94128700	2.01405200	H	0.90753700	2.62963900	1.90981600
C	-2.66133400	1.50069500	3.48154100	H	-3.06699800	0.06908800	1.32976600
C	0.69857600	1.99288000	-1.42067100	H	2.55583400	0.16056000	-1.50890100
C	0.96513400	2.77695000	0.83580100	H	2.35091400	-1.34386800	-2.41243600
C	-1.66581000	-1.28556600	-1.45114600	H	3.80114700	-2.70218600	-0.95708700
C	-2.97755100	-0.45933300	0.38817700	H	2.56944700	-3.42897200	3.09834900
C	1.97348500	-0.76750600	-1.55204300	H	-0.75079600	0.92362900	-2.60167900
C	3.14444900	-2.48800000	-0.11578900	H	0.86000200	0.96336400	-3.33094900
C	2.45576600	-2.89721400	2.15703900	H	1.30895200	3.35705300	-2.97050300
C	0.32090500	0.87829600	-2.37311100	H	1.78565000	4.75693500	1.06749700
C	1.20066900	3.20480100	-1.89820500	H	0.12993600	-2.42240300	-1.74273300
C	1.47093200	3.99049600	0.36384300	H	-0.40291400	-1.57758100	-3.20036000
C	-0.31656200	-1.48838400	-2.10493000	H	-2.75318200	-2.31215800	-2.99974200
C	-2.82221900	-1.79520000	-2.04422400	H	-5.09592000	-0.84025900	0.29253100

C	-0.01841000	-1.99129400	2.88041000	C	3.71550000	0.29175400	-1.30441700
H	0.07978900	-2.10541500	0.75809400	C	2.36752300	2.24767000	-0.23597200
C	-3.68500800	0.16866700	2.26952700	O	3.51104600	0.58379800	1.03349300
C	-1.96263200	-0.89218300	3.81094800	C	4.25884500	-1.11050600	-1.07556300
C	-1.63650300	2.47112700	-0.78697000	C	1.85507300	2.83801000	0.93351200
C	-3.91643900	1.96583900	-0.20429100	C	2.42666500	3.02786500	-1.39847000
C	-4.07478700	-3.01958400	-3.12688300	B	3.10445800	-0.22516700	2.20138800
H	-2.32268300	-1.80797100	-3.10700900	C	5.32816900	-1.32084600	-0.19777300
C	-4.82424500	-1.48431600	0.80612500	C	3.66796300	-2.21472500	-1.70037500
C	-5.47631300	-2.93740300	-1.16625700	C	1.41113400	4.15832800	0.93244800
C	-0.75042000	-1.56268500	3.98697500	H	1.81526300	2.24422000	1.84010900
H	0.94899800	-2.46563200	2.99364200	C	1.99155400	4.35874200	-1.39583900
N	-4.13943000	-0.16853400	0.88717400	H	2.80439600	2.59931100	-2.31767300
H	-3.51272100	1.24998100	2.29863800	F	2.84437200	-1.57560800	1.83666300
H	-4.47618400	-0.06413500	2.99324200	F	1.88805700	0.29577400	2.75297200
H	-2.54788400	-0.58110300	4.67379700	F	4.12772500	-0.15462400	3.12414600
C	-1.96280100	3.82600300	-0.87564000	C	5.79290700	-2.61014000	0.05216000
H	-0.60117400	2.19556300	-0.92620200	H	5.75821900	-0.47379800	0.32507500
C	-4.91491700	0.92566600	0.24155800	C	4.13846000	-3.50667000	-1.45734400
C	-4.25055300	3.31551300	-0.29967500	H	2.83426100	-2.05443600	-2.37808400
C	-5.21501100	-3.43884600	-2.44096800	C	1.48044800	4.93041000	-0.23192700
H	-3.86790800	-3.39986300	-4.12321900	H	1.01851700	4.59127700	1.84941100
H	-4.36543400	-2.13677100	1.55693600	H	2.06050600	4.95458200	-2.30907600
H	-5.88771100	-1.37984600	1.05472700	C	5.20326400	-3.70785200	-0.57916700
H	-6.36653800	-3.25148000	-0.62589900	H	6.61129900	-2.75870500	0.75146800
H	-0.37555000	-1.74729000	4.98931300	H	3.67005200	-4.35433600	-1.95237100
C	-3.27311600	4.24782000	-0.65159500	H	1.14795200	5.96604600	-0.22708100
H	-1.17717400	4.54103400	-1.09704500	H	5.56841000	-4.71289400	-0.38286500
H	-5.44783800	0.48841700	-0.61032400	C	3.77515300	0.37273600	-3.69489300
H	-5.66364600	1.32500800	0.93681000	H	3.09500200	0.40366600	-4.55112200
H	-5.26515200	3.64066900	-0.07994200	H	4.42620300	1.26059200	-3.72117000
H	-5.90092600	-4.14617000	-2.89844800	H	4.40206000	-0.52574100	-3.76228900
H	-3.52859700	5.30169500	-0.71921200	O	2.97064200	0.36521900	-2.52702400
C	-0.47577000	-0.71757600	-1.63788200	H	4.55014600	1.00801600	-1.33807700

Table S22. Optimized Cartesian coordinates (Å) for **Int(syn-3a)** in the singlet state at the B3LYP-D3/6-31G* level.

C	1.23856800	-1.83337100	2.96244600	H	-2.32851800	-1.13616900	1.79565500
C	0.26204900	-1.47293500	2.12941400	F	-0.80244100	3.80346500	0.06078900
H	1.34188800	-2.85849000	3.30816300	F	-1.89150200	2.33858700	1.42594600
H	1.97623500	-1.12004800	3.32477300	C	3.84231400	1.45149400	-0.98478800
C	0.04342400	-0.05109400	1.66809600	H	1.82207200	1.75344100	-1.66820100
H	-0.43100100	-2.22608400	1.76654900	C	4.25999800	-0.62249600	0.17697700
H	0.97040300	0.51661800	1.79841600	H	2.55303000	-1.92561200	0.40424800
H	-0.71010900	0.42993100	2.30379700	C	-4.50149700	-0.89771000	-0.79721300
C	-0.40089300	0.16069900	0.20399000	H	-4.06647100	0.03889600	-2.69254100
C	0.56804700	-0.51253100	-0.82299000	H	-4.62627200	-1.72211100	1.19024700
C	-1.84576800	-0.24182000	-0.11213900	C	4.72406800	0.59721300	-0.31847100
O	-0.25282300	1.57796700	-0.09683800	H	4.19557200	2.40070500	-1.37920000
C	2.03160200	-0.12989500	-0.64637600	H	4.94011900	-1.29619200	0.69203700
C	-2.37245300	0.09241200	-1.37020300	H	-5.52431500	-1.15326400	-1.06020300
C	-2.68273300	-0.88909200	0.80238600	H	5.76616100	0.87870500	-0.19141800
B	-0.98948300	2.55349600	0.46829800	C	0.70351000	-2.62973300	-1.88693900
C	2.50664600	1.08679400	-1.15284800	H	0.49076000	-3.68184400	-1.68205400
C	2.92174600	-0.98423300	0.01416400	H	0.10682100	-2.30374500	-2.75288100
C	-3.68309600	-0.23268700	-1.71255200	H	1.76992700	-2.51613200	-2.12655600
H	-1.75585500	0.63152300	-2.08337500	O	0.34576300	-1.90704000	-0.72113800
C	-3.99649100	-1.21878300	0.46155900	H	0.24469800	-0.15690500	-1.81283600

Table S23. Optimized Cartesian coordinates (Å) for **1SiF** in the singlet state at the B3LYP-D3/6-31G* level.

Si	0.00002100	-0.00002200	0.80493100	H	2.12845100	1.32150500	2.48351600
C	-1.77445900	0.55017200	0.47345100	N	-0.00011100	0.00009500	-1.48930600
C	0.41081900	-1.81180400	0.47338700	H	-1.57284800	-1.31443900	-1.84747600
C	1.36383100	1.26150900	0.47333400	H	-1.61876900	0.11814800	-2.88320000
C	-2.29486000	0.43097900	-0.82838400	C	-4.38571800	1.41279700	-0.12162900
C	-2.59426600	1.09649000	1.47175700	H	-3.97256300	0.77191500	-2.14119400
C	0.77413400	-2.20275800	-0.82853700	H	-4.51197600	1.95479500	1.96185100
C	0.34769900	-2.79502900	1.47159700	H	1.92471500	-0.70460800	-1.84740400
C	1.52054300	1.77195000	-0.82845600	H	0.70710200	-1.46057800	-2.88329000
C	2.24711000	1.69807600	1.47151200	C	0.96917300	-4.50450400	-0.12204600
C	-1.40770500	-0.23042800	-1.86107200	H	1.31745700	-3.82605700	-2.14157400
C	-3.58755100	0.86334600	-1.12728300	H	0.56319900	-4.88502700	1.96145800
C	-3.89020800	1.52731800	1.17932800	H	-0.35231700	2.01940100	-1.84695200
H	-2.20893800	1.18166200	2.48383700	H	0.91106400	1.34320300	-2.88319000
C	0.90336700	-1.10362000	-1.86110800	C	3.41635300	3.09172300	-0.12186000
C	1.04589400	-3.53843600	-1.12760300	H	2.65440000	3.05464000	-2.14129100
C	0.62245100	-4.13274800	1.17899900	H	3.94936700	2.92965500	1.96146300
H	0.08145400	-2.50400100	2.48375400	H	-5.39130600	1.75352700	-0.35474400
C	0.50394700	1.33445200	-1.86094700	H	1.17677600	-5.54572400	-0.35530500
C	2.54125100	2.67535200	-1.12741300	H	4.21415900	3.79228700	-0.35501800
C	3.26809500	2.60506900	1.17903800	F	0.00018400	-0.00001000	2.46271800

Second round of the review process

I have reviewed the authors' response and the revised manuscript. While the use of second-order perturbation energy to explain nucleophilicity differences among allylic species of the same element (Table S6 in Response to Referees Letter) is reasonable, my central concern, as noted by another reviewer, regarding the unconvincing explanation for reaction selectivity remains unresolved.

Specifically, in Table 1, non-cage allylic Si/Sn compounds yield syn-type chelation products, whereas cage-type analogues afford anti-type non-chelation products. This product distribution cannot be rationalized solely by nucleophilicity, as the second-order perturbation energies (9.37 kcal/mol for 1Sn(allyl) vs. 5.61 kcal/mol for allylSiPh₃) do not align with the observed stereochemical outcomes. The reactivity shown in Scheme 1b further confirms that product selectivity is not governed exclusively by nucleophilicity.

Although the Group 14 metal-containing heteroatom cage derivatives developed here address the challenges associated with the anti-selective addition of α -oxy carbonyl compounds, elucidating the origin of anti-selective addition is essential to validate this new strategy. This reviewer strongly recommend that the authors provide conclusive computational evidence to explain the syn/anti-type chelation/non-chelation selectivity in the next revision.

We would like to clarify that the nucleophilicity estimated based on second-order perturbation energy is not suitable for explaining the stereoselectivity of allylations involving α -oxy ketones. In fact, we do not rely on nucleophilicity to account for the observed stereoselectivity. It is important to note here that the nucleophilicity of allylic reagents, which is derived from the electronic properties of the allyl moiety, does not have a direct connection to the stereoselectivity in their nucleophilic additions.

Instead, the stereoselectivity is determined by the transition-state conformations of α -oxy carbonyl compounds during the nucleophile attack. Based on seminal studies (for example, ref. 9), it has been empirically and theoretically accepted that α -oxy carbonyls induce *anti*-adducts in their allylations via a non-chelation pathway following the polar Felkin-Anh model. Reetz and Evans (see refs. 9, 31, and 33) have explained the *anti*-selectivity exhibited by α -oxy aldehydes due to their high electrophilicity. Given that a fundamental understanding of *syn/anti*-selectivity in nucleophilic additions to α -substituted carbonyls is well-established through abundant theoretical and experimental studies, we would like to argue that further computational evidence for our reactions is unnecessary.

In contrast to these pioneering studies, achieving *anti*-selective additions to α -oxy ketones—which are less reactive than the corresponding aldehydes as demonstrated by Reetz and Evans—remains elusive. Our manuscript addresses this critical issue in allylations. For the allylation of α -oxy ketones, it is essential to enhance either the electrophilicity of the ketone moiety or the nucleophilicity of the allylic reagent. However, both approaches encounter a significant and unavoidable challenge in achieving *anti*-selectivity. Traditionally, reagents with strong Lewis acidity are employed in these reactions; however, they can increase both the electrophilicity of the ketone group and the

nucleophilicity of the allylic reagent. Unfortunately, this often results in chelation-controlled additions that favor *syn*-selective outcomes.

To address this inherent issue and realize *anti*-selectivity in the allylation of α -oxy ketones, our goal is to harmonize high nucleophilicity with low Lewis acidity in an allylic reagent. This concept is central to our atrane-type reagent. To underscore this important point, we have added the following sentence to the main text.

that the molecular geometry of **1Si(allyl)** exhibits a nearly trigonal-bipyramidal structure with a five-coordinated silicon center. Most allylic reagents with high nucleophilicity feature a strong Lewis-acidic metal center, which facilitates the allylation of α -oxy ketones through a chelation pathway, resulting in *syn*-products.^{9,31,33} In contrast to the traditional methods for achieving *syn*-selective allylations, **1E(allyl)** enhances the nucleophilicity of the allylic group while simultaneously reducing the Lewis acidity of the metal center. This careful balance between high nucleophilicity and weak Lewis acidity in **1E(allyl)**, achieved through an atrane-type framework, allows for the distinctive *anti*-selective allylations of α -oxy ketones.

First round of the review process

The manuscript by Yasuda et al. developed an anti-selective alkenylation of α -oxy ketones using the previously reported Group-14 allyltranes (Chem Asian J. 2021, 16, 3118–3123). The authors employ computational investigations to compare the reactivity of atrane-type allyl nucleophiles with that of allylSiPh₃, suggesting that the former exhibits high nucleophilicity and low chelating ability. Despite the new approach to anti-selective allylation via a non-chelation pathway, the study is not acceptable for publication due to fundamental issues that remain unclear. Firstly, the reported second-order perturbation energy ($\sigma(\text{Sn}-\text{C}\alpha) \rightarrow \pi^*(\text{C}\beta=\text{C}\gamma)$) is 9.37 kcal/mol for Sn(allyl) versus 5.61 kcal/mol for allylSiPh₃. However, a higher value indicates a stronger interaction, which is irreconcilable with the claim that transannular interactions stabilize the allyl-anion character. This contradiction invalidates the design of nucleophiles.

We are concerned that the reviewers may have misunderstood our findings. The extent of hyperconjugation, denoted as $\sigma(\text{E}-\text{C}\alpha) \rightarrow \pi^*(\text{C}=\text{C})$, is more pronounced when the atomic number of element E increases. Consequently, when comparing homologs that contain identical substituents or ligands, the second-order perturbation energy tends to increase in the order of Si < Ge < Sn. However, the reviewer's point is that the comparison is between totally different allylic species with different frameworks. Thus, the pointed comparison is not suitable for discussion.

We have calculated the value for allylSnPh₃ to be 9.53 kcal/mol, which is higher than that for 1Sn(allyl) (9.37 kcal/mol). This estimation also supports our arguments. We have added the calculated values for allylSnPh₃ to Table S6, Figure S14, and Figure S15.

Table S6. Summary of parameters related to nucleophilicity.

	1Si(allyl)	1Ge(allyl)	1Sn(allyl)	allylSiPh ₃	allylSi(OC ₂ H ₄) ₃ N	allylSnPh ₃
HOMO /eV	-6.18	-6.02	-5.91	-6.60	-6.17	-6.37
NBO charge						
q(E)	+1.891	+1.697	+1.898	+1.872	+2.379	+1.852
q(C α)	-1.038	-0.970	-0.986	-1.009	-1.061	-0.958
q(C β)	-0.215	-0.222	-0.230	-0.227	-0.217	-0.244
q(C γ)	-0.492	-0.491	-0.506	-0.481	-0.491	-0.491
$\sigma(\text{E}-\text{C}\alpha) \rightarrow \pi^*(\text{C}\beta=\text{C}\gamma)$ /kcal/mol	3.84	5.29	9.37	5.61	5.90	9.53
$\Delta\delta_{\text{C}\beta-\text{C}\gamma}$ /ppm	25.3	24.4	28.4	18.7	27.8	23.5

B3PW91/DGDZVP (for Si, Ge, and Sn), 6-31+G**(for C, H, and N) level.

Figure S14. HOMO of (a) 1Si(allyl), (b) 1Ge(allyl), (c) 1Sn(allyl), (d) allylSiPh₃, (e) allylSi(OC₂H₄)₃N, and (f) allylSnPh₃ calculated at the B3PW91/DGDZVP (for Si, Ge, and Sn), 6-31+G**(for C, H, and N) level.

Figure S15. Natural bond orbitals of (a) 1Si(allyl), (b) 1Ge(allyl), (c) 1Sn(allyl), (d) allylSiPh₃, (e) allylSi(OC₂H₄)₃N, and (f) allylSnPh₃ calculated at the B3PW91/DGDZVP (for Si, Ge, and Sn), 6-31+G**(for C, H, and N) level.

More importantly, the origin of the anti-selectivity remains obscure. This is highlighted by the observation in Scheme 1b that both nucleophiles (allylSiMe₃ and Si(allyl)) yield the anti-selective product with cyclobutanone, suggesting the stereochemical outcome is not uniquely dictated by the nucleophile's properties as claimed. The steric effect can be a pivotal factor that affects the anti-selectivity. Consequently, without deeper mechanistic investigations, this work remains descriptive and fails to provide a coherent or convincing explanation for the observed reactivity and selectivity.

The issue that the reviewer pointed out is scientifically significant. We are very sorry for any potentially unclear discussions on this point, because the mechanism for the stereoselectivity in the allylation of cyclic ketones is not directly explained based on the Felkin-Anh model. In these reactions, the main interest is the steric demand of the allylsilatrane. To clarify this fact, we have modified the corresponding sentence in Scheme 1b, as shown below.

Furthermore, the steric effect of **1**Si(allyl) was evident in the allylation of α -oxy cyclobutanone **2q** (Scheme 1b). The resulting allylated product (**3q**) is an important synthetic intermediate for the substructure of a β -secretase modulator candidate.